# ILLITERATE DALL-E LEARNS TO COMPOSE

**Gautam Singh**[1]**, Fei Deng**[1] **& Sungjin Ahn**[2]
[1]Rutgers University
[2]KAIST

## ABSTRACT

Although DALL·E has shown an impressive ability of composition-based systematic generalization in image generation, it requires the dataset of text-image pairs and the compositionality is provided by the text. In contrast, object-centric representation models like the Slot Attention model learn composable representations without the text prompt. However, unlike DALL·E its ability to systematically generalize for zero-shot generation is significantly limited. In this paper, we propose a simple but novel slot-based autoencoding architecture, called SLATE[1], for combining the best of both worlds: learning object-centric representations that allows systematic generalization in zero-shot image generation without text. As such, this model can also be seen as an illiterate DALL·E model. Unlike the pixel-mixture decoders of existing object-centric representation models, we propose to use the Image GPT decoder conditioned on the slots for capturing complex interactions among the slots and pixels. In experiments, we show that this simple and easy-to-implement architecture not requiring a text prompt achieves significant improvement in in-distribution and out-of-distribution (zero-shot) image generation and qualitatively comparable or better slot-attention structure than the models based on mixture decoders. https://sites.google.com/view/slate-autoencoder

## 1  INTRODUCTION

Unsupervised learning of compositional representation is a core ability of human intelligence (Yuille & Kersten, 2006; Frankland & Greene, 2020). Observing a visual scene, we perceive it not simply as a monolithic entity but as a geometric composition of key components such as objects, borders, and space (Kulkarni et al., 2015; Yuille & Kersten, 2006; Epstein et al., 2017; Behrens et al., 2018). Furthermore, this understanding of the structure of the scene composition enables the ability for *zero-shot imagination*, i.e., composing a novel, counterfactual, or systematically manipulated scenes that are significantly different from the training distribution. As such, realizing this ability has been considered a core challenge in building a human-like AI system (Lake et al., 2017).

DALL·E (Ramesh et al., 2021) has recently shown an impressive ability to systematically generalize for zero-shot image generation. Trained with a dataset of text-image pairs, it can generate plausible images even from an unfamiliar text prompt such as "avocado chair" or "lettuce hedgehog", a form of systematic generalization in the text-to-image domain (Lake & Baroni, 2018; Bahdanau et al., 2018). However, from the perspective of compositionality, this success is somewhat expected because the text prompt already brings the composable structure. That is, the text is already discretized into a sequence of concept modules, i.e., words, which are composable, reusable, and encapsulated. DALL·E then learns to produce an image with global consistency by smoothly stitching over the discrete concepts via an Image GPT (Chen et al., 2020) decoder.

Building from the success of DALL·E, an important question is if we can achieve such systematic generalization for zero-shot image generation only from images without text. This would require realizing an ability lacking in DALL·E: extracting a set of composable representations from an image to play a similar role as that of word tokens. The most relevant direction towards this is object-centric representation learning (Greff et al., 2019; Locatello et al., 2020; Lin et al., 2020b;

---

Correspondence to singh.gautam@rutgers.edu and sjn.ahn@gmail.com.
[1]The implementation is available at https://github.com/singhgautam/slate.

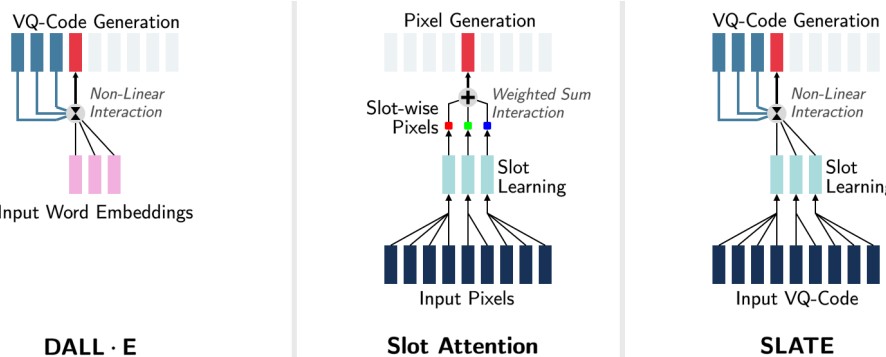

**Figure 1: Overview of our proposed model with respect to prior works. Left:** In DALL·E, words in the input text act as the composable units for generating the desired novel image. The generated images have global consistency because each pixel depends non-linearly on all previous pixels and the input word embeddings because of its transformer decoder. **Middle:** Unlike DALL·E that requires text supervision, Slot Attention provides an auto-encoding framework in which object slots act as the composable units inferred purely from raw images. However, during rendering, object slots are composed via a simple weighted sum of pixels obtained without any dependency on other pixels and slots which harms image consistency and quality. **Right:** Our model, SLATE, combines the best of both models. Like Slot Attention our model is free of text-based supervision and like DALL·E, produces novel image compositions with global consistency.

Greff et al., 2020b). While this approach can obtain a set of slots from an image by reconstructing the image from the slots, we argue that its ability to systematically generalize in such a way to handle arbitrary *reconfiguration* of slots is significantly limited due to the way it decodes the slots.

In this work, we propose a slot-based autoencoder architecture, called SLot Attention TransformEr (SLATE). SLATE combines the best of DALL·E and object-centric representation learning. That is, compared to the previous object-centric representation learning, our model significantly improves the ability of composition-based systematic generalization in image generation. However, unlike DALL·E we achieve this by learning object-centric representations from images alone instead of taking a text prompt, resulting in a text-free DALL·E model. To do this, we first analyze that the existing pixel-mixture decoders for learning object-centric representations suffer from the *slot-decoding dilemma* and the *pixel independence problem* and thus show limitations in achieving compositional systematic generalization. Inspired from DALL·E, we then hypothesize that resolving these limitations requires not only learning composable slots but also a powerful and expressive decoder that can model complex interactions among the slots and the pixels. Based on this investigation, we propose the SLATE architecture, a simple but novel slot autoencoding architecture that uses a slot-conditioned Image GPT decoder. Also, we propose a method to build a library of visual concepts from the learned slots. Similar to the text prompt in DALL·E, this allows us to program a form of 'sentence' made up of visual concepts to compose an image. Furthermore, the proposed model is also simple, easy to implement, can be plugged in into many other models. In experiments, we show that this simple architecture achieves various benefits such as significantly improved generation quality in both the in-distribution and the out-of-distribution settings. This suggests that more attention and investigation on the proposed framework is required in future object-centric models.

The main contribution of the paper can be seen as four folds: (1) achieving the first text-free DALL·E model, (2) the first object-centric representation learning model based on a transformer decoder, and (3) showing that it significantly improves the systematic generalization ability of object-centric representation models. (4) While achieving better performance, the proposed model is much simpler than previous approaches.

## 2 PRELIMINARIES

### 2.1 OBJECT-CENTRIC REPRESENTATION LEARNING WITH PIXEL-MIXTURE DECODER

A common framework for learning object-centric representations is via a form of auto-encoders (Locatello et al., 2020; Burgess et al., 2019; Lin et al., 2020b). In this framework, an encoder takes an input image to return a *set* of object representation vectors or slot vectors $\{\mathbf{s}_1, \ldots, \mathbf{s}_N\} = f_\phi(\mathbf{x})$. A decoder then reconstructs the image by composing the decodings $g_\theta(\mathbf{s}_n)$ of each slot, $\hat{\mathbf{x}} = f_{\text{compose}}(g_\theta(\mathbf{s}_1), \ldots, g_\theta(\mathbf{s}_N))$. To encourage the emergence of object concepts in the slots, the decoder usually uses an architecture implementing an inductive bias about the scene composition such as the pixel-mixture decoders.

**Pixel-mixture decoders** are the most common approach to construct an image from slots. This method assumes that the final image is constructed by the pixel-wise weighted mean of the slot images. Specifically, each slot $n$ generates its own slot image $\boldsymbol{\mu}_n$ and its corresponding alpha-mask $\boldsymbol{\pi}_n$. Then, the slot images are weighted-summed with the alpha-mask weights as follows:

$$\hat{\mathbf{x}}_i = \sum_{n=1}^{N} \pi_{n,i} \cdot \boldsymbol{\mu}_{n,i} \quad \text{where} \quad \boldsymbol{\mu}_n = g_\theta^{\text{RGB}}(\mathbf{s}_n) \quad \text{and} \quad \boldsymbol{\pi}_n = \frac{\exp g_\theta^{\text{mask}}(\mathbf{s}_n)}{\sum_{m=1}^{N} \exp g_\theta^{\text{mask}}(\mathbf{s}_m)},$$

where $i \in [1, HW]$ is the pixel index with $H$ and $W$ the image height and width, respectively.

### 2.2 LIMITATIONS OF PIXEL-MIXTURE DECODERS

The pixel-mixture decoder has two main limitations. The first is what we term as the **slot-decoding dilemma**. As there is no explicit incentive for the encoder to make a slot focus on the pixel area corresponding to an object, it is usually required to employ a weak decoder for $g_\theta^{\text{RGB}}$ such as the Spatial Broadcast Network (Watters et al., 2019) to make object concepts emerge in the slots. By limiting the capacity of the slot-decoder, it prevents a slot from modeling multiple objects or the entire image and instead incentivizes the slot to focus on a local area that shows a statistical regularity such as an object of a simple color. However, this comes with a side effect: the weak decoder can make the generated image blur the details. When we use a more expressive decoder such as a CNN to prevent this, the object disentanglement tends to fail, e.g., by producing slots that capture the whole image, hence a dilemma (See Appendix E.1).

The second problem is the **pixel independence**. In pixel-mixture decoders, each slot's contribution to a generated pixel, i.e., $\pi_{n,i}$ and $\boldsymbol{\mu}_{n,i}$, is independent of the other slots and pixels. This may not be an issue if the aim of the slots is to reconstruct only the input image. However, it becomes an issue when we consider that an important desired property for object representations is to use them as *concept modules* i.e., to be able to *reuse* them in an arbitrary configuration in the same way as word embeddings can be reused to compose an arbitrary image in DALL·E. The lack of flexibility of the decoder due to the pixel independence, however, prevents the slots from having such reconfigurability. For example, when decoding an arbitrary set of slots taken from different images, the rendered image would look like a mere superposition of individual object patches without *global semantic consistency*, producing a Frankenstein-like image as the examples in Section 5 shall show. In contrast, this zero-shot global semantic consistency for concept reconfiguration is achieved in DALL·E by using the Image GPT decoder.

### 2.3 IMAGE GPT AND DALL·E

**Image GPT** (Chen et al., 2020a) is an autoregressive generative model for images implemented using a transformer (Vaswani et al., 2017; Brown et al., 2020). For computational efficiency, Image GPT first down-scales the image of size $H \times W$ by a factor of $K$ using a VQ-VAE encoder (van den Oord et al., 2017). This turns an image $\mathbf{x}$ into a sequence of discrete image tokens $\{\mathbf{z}_i\}$ where $i$ indexes the tokens in a raster-scan order from 1 to $HW/K^2$. The transformer is then trained to model an auto-regressive distribution over this token sequence by maximizing the log-likelihood $\sum_i \log p_\theta(\mathbf{z}_i|\mathbf{z}_{<i})$. During generation, the transformer samples the image tokens sequentially conditioning on the previously generated tokens, i.e., $\hat{\mathbf{z}}_i \sim p_\theta(\hat{\mathbf{z}}_i|\hat{\mathbf{z}}_{<i})$ where $\hat{\mathbf{z}}_i$ is the image token generated for the position $i$ in the sequence. Lastly, a VQ-VAE decoder takes all the generated tokens and produces the final image $\hat{\mathbf{x}}$. Crucially, due to the autoregressive generation of $\mathbf{z}_i$, unlike

the pixel-mixture decoder, the generated pixels are not independent. The autoregressive generation powered by the direct context attention of the transformer makes Image GPT one of the most powerful decoder models in image generation and completion tasks. However, Image GPT does not provide a way to learn high-level semantic representations such as object-centric representations that can provide a way to semantically control image generation.

**DALL·E** (Ramesh et al., 2021) shows that the image generation of Image GPT can be controlled by conditioning on a text prompt. In particular, it shows the impressive ability of zero-shot generation *with global semantic consistency* for rather an arbitrary reconfiguration of the words in the text prompt. DALL·E can be seen as a conditional Image GPT model, $\hat{\mathbf{z}}_i \sim p_\theta(\hat{\mathbf{z}}_i|\hat{\mathbf{z}}_{<i}, \mathbf{c}_{1:N})$, where $\mathbf{c}_{1:N}$ are the text prompt tokens. The key limitation from the perspective of this work is that it requires supervision in terms of text-image pairs. This means that it does not need to learn the compositional structure by itself since the discrete concepts are inherently provided by the prompt text even though it learns the word embeddings provided the structure. Therefore, it is important to investigate if we can make a text-free DALL·E by learning to infer such compositional structure and representation only from images.

## 3 SLATE: SLOT ATTENTION TRANSFORMER

In this section, we propose a method that can provide global semantic consistency in the zero-shot image generation setting without text by learning object slots from images alone. In doing this, we shall lift the use of inductive biases in the image decoder about slot composition by making use of a transformer (Vaswani et al., 2017) as our image decoder while replacing the text prompt with a slot prompt drawn from a concept library constructed from a given set of images. Our central hypothesis is that a powerful auto-regressive decoder such as a transformer after training with a sufficiently diverse set of raw images should learn the slot representations and the rules of composition jointly in a way to achieve systematic generalization of zero-shot image generation. In the following, we describe the details of the model architecture.

**Obtaining Image Tokens using DVAE.** To make the training of transformer computationally feasible for high-resolution images, we first downscale the input image $\mathbf{x}$ of size $H \times W$ by a factor of $K$ using Discrete VAE (Im et al., 2017). To do this, we split the image into $K \times K$-sized patches resulting in $T$ patches where $T = HW/K^2$. We provide each patch $\mathbf{x}_i$ as input to an encoder network $f_\phi$ to return log probabilities (denoted as $\mathbf{o}_i$) for a categorical distribution with $V$ classes. With these log probabilities, we use a relaxed categorical distribution (Jang et al., 2016) with a temperature $\tau$ to sample a relaxed one-hot encoding $\mathbf{z}_i^{\text{soft}}$ for the patch $i$. We then decode these soft vectors to obtain patch reconstructions. This process is summarized as follows:

$$\mathbf{o}_i = f_\phi(\mathbf{x}_i) \quad \Longrightarrow \quad \mathbf{z}_i^{\text{soft}} \sim \text{RelaxedCategorical}(\mathbf{o}_i; \tau) \quad \Longrightarrow \quad \tilde{\mathbf{x}}_i = g_\theta(\mathbf{z}_i^{\text{soft}}) .$$

By minimizing an MSE reconstruction objective for the image patches i.e. $\mathcal{L}_{\text{DVAE}} = \sum_{i=1}^{T}(\tilde{\mathbf{x}}_i - \mathbf{x}_i)^2$, we train the DVAE encoder $f_\phi(\cdot)$ and the decoder $g_\theta(\cdot)$ networks.

**Inferring Object Slots.** To infer the object slots from a given image $\mathbf{x}$, we first use the DVAE encoder as described above to obtain a discrete token $\mathbf{z}_i$ for each patch $i$. Next, we map each token $\mathbf{z}_i$ to an embedding by using a learned dictionary. To incorporate the position information into these patch embeddings, we sum them with learned positional embeddings. This results in an embedding $\mathbf{u}_i$ which now has both the content and the position information for the patch. These embeddings $\mathbf{u}_{1:T}$ are then given as input to a Slot-Attention encoder (Locatello et al., 2020) with $N$ slots.

$$\mathbf{o}_i = f_\phi(\mathbf{x}_i) \quad \Longrightarrow \quad \mathbf{z}_i \sim \text{Categorical}(\mathbf{o}_i) \quad \Longrightarrow$$
$$\mathbf{u}_i = \text{Dictionary}_\phi(\mathbf{z}_i) + \mathbf{p}_{\phi,i} \quad \Longrightarrow \quad \mathbf{s}_{1:N}, A_{1:N} = \text{SlotAttention}_\phi(\mathbf{u}_{1:T}) .$$

This results in $N$ object slots $\mathbf{s}_{1:N}$ and $N$ attention maps $A_{1:N}$ from the last refinement iteration of the Slot-Attention encoder.

**Reconstruction using Transformer.** To reconstruct the input image, we decode the slots $\mathbf{s}_{1:N}$ and first reconstruct the DVAE tokens $\hat{\mathbf{z}}_{1:T}$ using a transformer. We then use the DVAE decoder $g_\theta$ to decode the DVAE tokens and reconstruct the image patches $\hat{\mathbf{x}}_i$ and therefore the image $\hat{\mathbf{x}}$.

$$\hat{\mathbf{o}}_i = \text{Transformer}_\theta(\hat{\mathbf{u}}_{<i}; \mathbf{s}_{1:N}) \quad \Longrightarrow \quad \hat{\mathbf{z}}_i = \arg\max_{v \in [1,V]} \hat{o}_{i,v} \quad \Longrightarrow \quad \hat{\mathbf{x}}_i = g_\theta(\hat{\mathbf{z}}_i) .$$

where $\hat{\mathbf{u}}_i = \text{Dictionary}_\phi(\hat{\mathbf{z}}_i) + \mathbf{p}_{\phi,i}$ and $\hat{o}_{i,v}$ is the log probability of token $v$ among the $V$ classes of the categorical distribution represented by $\hat{\mathbf{o}}_i$. To train the transformer, the slot attention encoder and the embeddings, we minimize the cross-entropy of predicting each token $\mathbf{z}_i$ conditioned on the preceding tokens $\mathbf{z}_{<i}$ and the slots $\mathbf{s}_{1:N}$. Let the predicted log-probabilities for the token at position $i$ be $\bar{\mathbf{o}}_i = \text{Transformer}_\theta(\mathbf{u}_{<i}; \mathbf{s}_{1:N})$. Then the cross-entropy training objective can be written as $\mathcal{L}_{\text{ST}} = \sum_{i=1}^T \text{CrossEntropy}(\mathbf{z}_i, \bar{\mathbf{o}}_i)$.

**Learning Objective and Training.** The complete training objective is given by $\mathcal{L} = \mathcal{L}_{\text{ST}} + \mathcal{L}_{\text{DVAE}}$ and all modules of our model are trained jointly. At the start of the training, we apply a decay on the DVAE temperature $\tau$ from 1.0 to 0.1 and a learning rate warm-up for the parameters of slot-attention encoder and the transformer.

## 3.1 VISUAL CONCEPT LIBRARY

The above model enables extracting a set of concepts or slots from a particular image whereas an intelligent agent would build a *library of reusable concepts* from diverse experience such as a stream of images. In DALL·E, a vocabulary of word embeddings plays the role of this library and provides reusable concepts. Similarly, to build a library of visual concepts from the dataset, we use the following simple approach based on $K$-means clustering to construct a library of reusable visual concepts. To do this, we propose the following steps: (i) Collect $N$ slots $\mathcal{S}_i = \{\mathbf{s}_1^i, \ldots, \mathbf{s}_N^i\}$ and their attention maps $\mathcal{A}_i = \{A_1^i, \ldots, A_N^i\}$ from each image $i$ in the training dataset. (ii) Run $K$-means clustering on $\mathcal{S} = \bigcup_i \mathcal{S}_i$ with cosine similarity between slots as a distance metric where $K$ is the number of total concepts in the library. If the object position is more important for building the concept library, cluster the slots using IOU between the attention maps as a similarity metric. (iii) Use each cluster as a concept and the slots assigned to the cluster as the instantiations of the concept. (iv) To compose an arbitrary image, choose the concepts from the library and randomly choose a slot for each concept and build a *slot prompt* similar to a text prompt in DALL·E. (v) Provide this prompt to the decoder to compose and generate the image.

## 4 RELATED WORK

**Compositional Generation via Object-Centric Learning.** Self-supervised object-centric approaches (Burgess et al., 2019; Greff et al., 2019; Locatello et al., 2020; Kabra et al., 2021; Greff et al., 2017; Engelcke et al., 2020; 2021; Eslami et al., 2016; Crawford & Pineau, 2019b; Lin et al., 2020b; Jiang & Ahn, 2020; Kosiorek et al., 2018; Jiang et al., 2019; Crawford & Pineau, 2019a; Lin et al., 2020a; Deng et al., 2021; Anciukevicius et al., 2020; von Kügelgen et al., 2020; Wu et al., 2021) typically perform mixture-based alpha compositing for rendering with contents of each slot decoded independently. Savarese et al. (2021) and Yang et al. (2020) rely on minimizing mutual information between predicted object segments as a learning signal while Yang et al. (2021) leverage optical flow. However, these approaches either cannot compose novel scenes or require specialized losses for object discovery unlike ours which uses a simple reconstruction loss. DINO (Caron et al., 2021) combining Vision Transformer (Dosovitskiy et al., 2021) and a self-supervised representation learning objective (Chen et al., 2020b; He et al., 2020; Grill et al., 2020; Touvron et al., 2021; Hinton, 2021; van den Oord et al., 2018) discovers object-centric attention maps but, unlike ours, cannot compose novel scenes. TIMs (Lamb et al., 2021; Goyal et al., 2021) with an Image GPT decoder show object-centric attention maps but, unlike ours, do not investigate compositional generation.

**Compositional Generation using GANs.** Chai et al. (2021) show compositional generation using a collage of patches encoded and decoded using a pretrained GAN. However, the collage needs to be provided manually and object discovery requires a specialized process unlike our model. Bielski & Favaro (2019) and Chen et al. (2019) show object discovery from given images by randomly re-drawing or adding a random jitter (Voynov et al., 2021) to the foreground followed by an adversarial loss. As these models use alpha compositing, their compositional ability is limited. Some works (Donahue et al., 2016; Donahue & Simonyan, 2019) infer abstract representations for images that emphasize the high-level semantics, but these, unlike ours, cannot be used for compositional image generation. Niemeyer & Geiger (2021), Nguyen-Phuoc et al. (2020); Chen et al. (2016), van Steenkiste et al. (2020), Liao et al. (2020) and Ehrhardt et al. (2020) introduce GANs which generate images conditioned on object-wise or factor-wise noise and optionally on camera pose. Lacking an inference module, these cannot perform compositional editing of a given image unlike ours. While

| Dataset | FID (↓) | | Human (↑) |
| --- | --- | --- | --- |
| | SA | Ours | Favor Ours |
| 3D Shapes | 44.14 | **36.75** | 50.36% |
| CLEVR | 63.76 | **41.62** | 84.66% |
| Shapestacks | 155.74 | **51.27** | 78.00% |
| Bitmoji | 71.43 | **15.83** | 95.06% |
| TexMNIST | 245.09 | **64.15** | 84.28% |
| CelebA | 179.07 | **62.72** | 97.34% |
| CLEVRTex | 195.82 | **93.76** | 99.02% |

**(a)** Compositional Generation

| Dataset | MSE (↓) | | FID (↓) | |
| --- | --- | --- | --- | --- |
| | SA | Ours | SA | Ours |
| 3D Shapes | **47.94** | 71.79 | 58.84 | **58.59** |
| CLEVR | **29.59** | 49.95 | 55.95 | **37.42** |
| Shapestacks | 233.72 | **111.86** | 139.72 | **30.22** |
| Bitmoji | 388.72 | **261.10** | 67.66 | **11.89** |
| TexMNIST | 295.20 | **149.02** | 179.95 | **55.00** |
| CelebA | **722.62** | 1047.65 | 161.25 | **34.22** |
| CLEVRTex | 526.17 | **498.39** | 193.04 | **58.91** |

**(b)** Image Reconstruction

**Table 1:** Comparison of Compositional Generation (left) and Image Reconstruction (right) between Slot-Attention (SA) and our model. For comparison of compositional generation, we report the FID score and average vote percentage received in favor of the generated images from our model. Vote percentage higher than 50% implies that our model was preferred over the Slot Attention baseline. For image reconstruction quality, we report the MSE and the FID score.

Kwak & Zhang (2016) provide an encoder to discover objects, the decoding relies on alpha compositing. Reed et al. (2016a) show compositional generation in GANs and improved generation quality (Johnson et al., 2018; Hinz et al., 2019) via an object-centric or a key-point based pathway for scene rendering. However, these require supervision for the bounding boxes and the keypoints.

# 5 EXPERIMENTS

In experiments, we evaluate the benefits of SLATE having a transformer decoder over the traditional pixel-mixture decoder in: 1) generating novel images from arbitrary slot configurations, 2) image reconstruction, and 3) generating out-of-distribution images. As the baseline, we compare with an unsupervised object-centric framework having a slot attention encoder like ours but using a mixture decoder for decoding instead of transformer. Hereafter, we refer to this baseline as simply Slot Attention (Locatello et al., 2020). We evaluate the models on 7 datasets which contain composable objects: 3D Shapes (Burgess & Kim, 2018), CLEVR-Mirror which we develop from the CLEVR (Johnson et al., 2017) dataset by adding a mirror in the scene, Shapestacks (Groth et al., 2018), Bitmoji (Graux, 2021), Textured MNIST, CLEVRTex (Karazija et al., 2021) and CelebA. For brevity, we shall refer to CLEVR-Mirror as simply CLEVR. The models take only raw images without any other supervision or annotations.

## 5.1 COMPOSITIONAL IMAGE GENERATION

We evaluate how effectively the known slots can be reused to generate novel scenes. For this, we evaluate the quality of image generations produced by our model from arbitrarily composed slot prompts. To build these prompts, we first build a concept library using $K$-means as described in Section 3.1. This results in clusters that correspond to semantic concepts such as hair, face, digits, foreground object, floor or wall. These are visualized in Appendix D. To build prompts from this library, we randomly pick one slot from each cluster. When randomly selecting the slots in CLEVR and CLEVRTex, we ensure that objects are separated by a minimum distance while in Shapestacks, we ensure that the objects are in a tower configuration. For more details, see Appendix F. In this way, we generate 40000 prompts for each dataset and render the corresponding images. We then evaluate the quality of the images by computing the FID score with respect to the true images. To support this metric, we also report the training curves of a CNN discriminator that tries to classify between real and model-generated images. If generated images are close to the true data distribution, then it should be harder for the discriminator to discriminate and the resulting training curve should converge more slowly. We also perform a human evaluation by reporting percentage of users who prefer our generations compared to the baseline.

From Table 1a and Figure 2, we note that our compositional generations are significantly more realistic than the generations from the mixture decoder. The qualitative samples shown in Figures 3, 12, 5 and 6 show that our decoder can compose the desired image accurately and make an effective

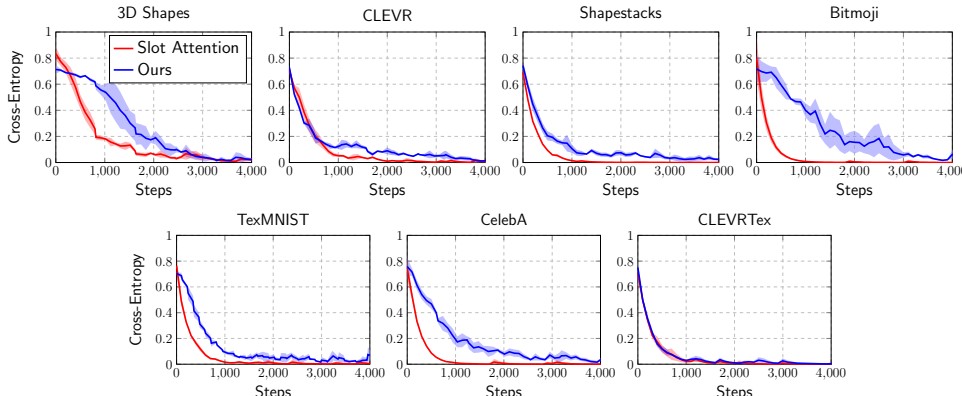

**Figure 2:** Comparison of training curves of a discriminator to compare the quality of compositional generation between Slot-Attention and our model. A CNN discriminator tries to discriminate between real images and model-generated images. We note that the curves converge more slowly for our model than for Slot Attention and show that our generations are more realistic.

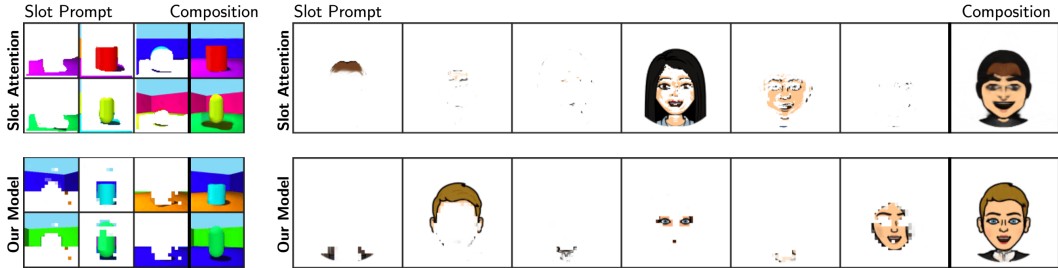

**Figure 3: Comparison of compositional generation between Slot Attention and our model in 3D Shapes and Bitmoji datasets.** We visualize the slot prompts provided to the decoder and the resulting compositions. To visualize the prompt slots, we show the image from which the slot was taken masked using its input attention map. Note that these attention maps had emerged in the source images implicitly without any supervision. We note that our decoder performs zero-shot generalization to novel slot inputs while the mixture decoder fails to compose arbitrary slots correctly.

reuse of the known slots. Additional qualitative results are provided in Appendix D. We also note that for visually complex datasets, the improvement in the FID score resulting from our model is much larger than that for the visually simpler CLEVR and 3D Shapes datasets.

Our results show that *i)* our model resolves the *slot-decoding dilemma* and consequently, it benefits from the flexibility of the Image GPT decoder that can render the fine details such as floor texture in Shapestacks and complex facial features in Bitmoji while the mixture decoder suffers. In Figures 4 and Figures 25 – 29, we also note that the attention masks of our model effectively focus on the object regions. This emergent object localization occurs despite the powerful decoder, thus resolving the dilemma. Qualitatively, our attention maps are comparable to those of Slot Attention and significantly better in case of textured images (such as CelebA and Textured MNIST) as shown in Figure 4. Because our slots attend on the DVAE tokens, hence our attention is patch-level and our attention segments may appear coarser than those of Slot Attention in which slots attend on the CNN feature cells having the same resolution as the input image. *ii)* Because the mixture decoder has the *pixel-independence assumption*, we see in Figure 3 that it suffers when slots are combined in arbitrary configurations. In 3D Shapes, the object shadows are controlled by the floor component of the mixture and are either too large or too small for the 3D object. This is because after arbitrarily choosing a different 3D object, the floor slot has no knowledge that the object which casts the shadow has changed and it continues to draw the shadow mask of the source image which has now become obsolete in the new slot configuration. Slot Attention suffers even more severely in Bitmoji as the composed mixture components and masks are mutually incompatible in the new slot configuration and produce an inaccurate alpha composition. In contrast, the SLATE decoder is robust to this

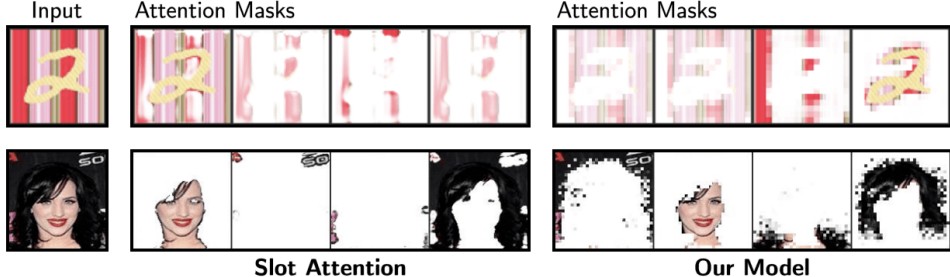

**Figure 4: Object Attention Masks in Textured Images.** We show qualitatively that our model produces better object attention masks in textured images whereas Slot Attention suffers. In Textured-MNIST, we note that the Slot Attention baseline fails to correctly segment the digit while our model succeeds. In CelebA (bottom row), we note that Slot Attention, unlike our model, incorrectly merges the black hair and the black background into the same slot due to their similar colors.

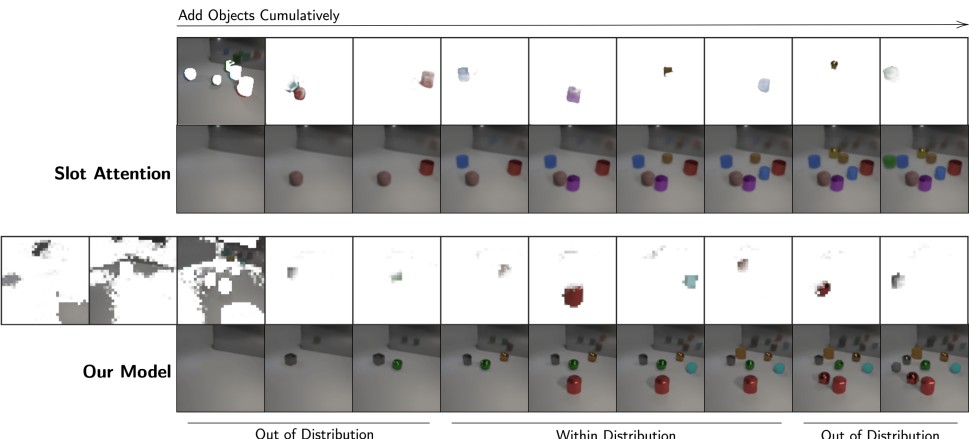

**Figure 5: Compositional Generation in CLEVR.** We show compositional generation in CLEVR by adding object slots one at a time. In training, only 3-6 objects were shown. However, we note that our model can generalize in generating 1-2 and 7-8 objects including the empty scene. We also note that our mirror reflections are significantly more clear than those of Slot Attention.

problem because the slots and pixels can interact. For instance, in 3D Shapes, as the pixels of the floor are being rendered, the predictor is aware of the object that will cast the shadow via the previous pixels and the input slots and can thus predict a globally consistent pixel value.

## 5.2 RECONSTRUCTION QUALITY

A natural test about whether an auto-encoder learns accurate representations of the given input is through an evaluation of the reconstruction error. For this, we report two metrics: 1) MSE to evaluate how well the reconstructed image preserves the contents of the original image and 2) FID score to show how realistic is the reconstructed image with respect to the true image distribution. We compute these metrics on a held-out set and report in Table 1b. We find that our model outperforms Slot Attention in all datasets in terms of FID which shows that the image quality of our model is better due to its powerful decoder. The FID improvement resulting from our model becomes larger as the visual complexity of the dataset increases. In MSE, our model performance is comparable to the mixture-based baseline. Despite the better reconstruction MSE of the baseline in 3D Shapes, CLEVR and CelebA, the gap is not severe considering that *i)* unlike the baseline, our model does not directly minimize MSE and *ii)* the baseline suffers significantly in rendering novel slot configurations as shown in Section 5.1.

**Figure 6: Compositional generation in Shapestacks Dataset.** We show compositional generation in Shapestacks by composing arbitrary blocks drawn from the concept library in a tower configuration. We note that SLATE can render the details such as the texture of the floor and the wall significantly better than the mixture decoder.

**Figure 7: Out-of-Distribution Generalization in Shapestacks Dataset.** We provide slot prompts to render two towers instead of one. The models were trained only on images with one tower.

| Dataset | FID ($\downarrow$) | |
|---|---|---|
| | SA | Ours |
| Shapestacks | 172.82 | **82.38** |
| CLEVR | 62.44 | **43.77** |
| Bitmoji | 81.66 | **22.75** |

**Table 2:** FID Comparison for OOD Compositions.

## 5.3 GENERALIZATION TO OUT-OF-DISTRIBUTION (OOD) SLOT RECONFIGURATIONS

Because the SLATE decoder is based on transformer and does not have an explicit inductive bias about scene composition, we test whether it can generalize in composing slots when their configurations are different from the training distribution. For this, *i)* in Shapestacks, we compose slot prompts to render two towers while only single-tower images were shown during training. *ii)* In CLEVR, we compose prompts to render fewer or more objects than were shown during training. *iii)* Lastly, in Bitmoji, we annotated the gender of 500 images and used it to construct 1600 OOD slot prompts by composing hair and face only from opposite genders. By doing this, we change the distribution of the slot configurations with respect to the training distribution. Note that such slot prompts were already a part of the Bitmoji compositions in Section 5.1. However, with this experiment, we evaluate the specific OOD subset of those prompts. We show the results in Figures 12, 5 and 7 and a comparison of FID score with respect to the mixture decoder in Table 2. From these, we note that our decoder is able to generalize to the OOD slot configurations and it also outperforms the mixture decoder in image quality.

## 6 DISCUSSION

We presented a new model, SLATE, for text-free zero-shot imagination by learning slot representations only from images. Our model combines the best of DALL·E and object-centric representation learning. It provides a significant improvement over pixel-mixture decoders in reconfigurability of slots for zero-shot imagination. These results suggest that future object-centric models may benefit from building on the advantages of the proposed framework alongside the pixel-mixture decoder.

The decoder of SLATE, with its transformer-based implementation, can also be seen as a graphics engine that learns to mimic the physical laws governing the rendering of the image through its exposure to a large dataset. Thus to render a scene, the only information that the transformer requires and requests from the encoder is the abstract object-level *what* and *where* in the form of slots. Knowing these objects, the transformer then fills in the rest and implicitly handles the mechanisms that give rise to the observed object appearance, occlusions, shadows, reflections, and transparency.

An interesting future direction is to consider that an intelligent agent may collect new experience indefinitely, and thus our approach of building the library of reusable concepts can be made more formal by learning discrete concepts and growing this library using online clustering (Liberty et al., 2016) or Bayesian non-parametric clustering (Neal, 2000). Another direction is to learn a slot-level prior for efficient density modeling and sampling scenes at the representation level.

## ETHICS STATEMENT

For future applications, negative societal consequences should also be considered. The current model does not generate images realistic enough to deceive humans. However, future versions of the model after scaling up its size and the training dataset can have such potential. Despite these considerations, these consequences are not imminent.

## ACKNOWLEDGEMENTS

This work is supported by Brain Pool Plus (BP+) Program (No. 2021H1D3A2A03103645) and Young Researcher Program (No. 2022R1C1C1009443) through the National Research Foundation of Korea (NRF) funded by the Ministry of Science and ICT.

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

# A  IMPLEMENTATION OF SLATE

## A.1  ARCHITECTURE

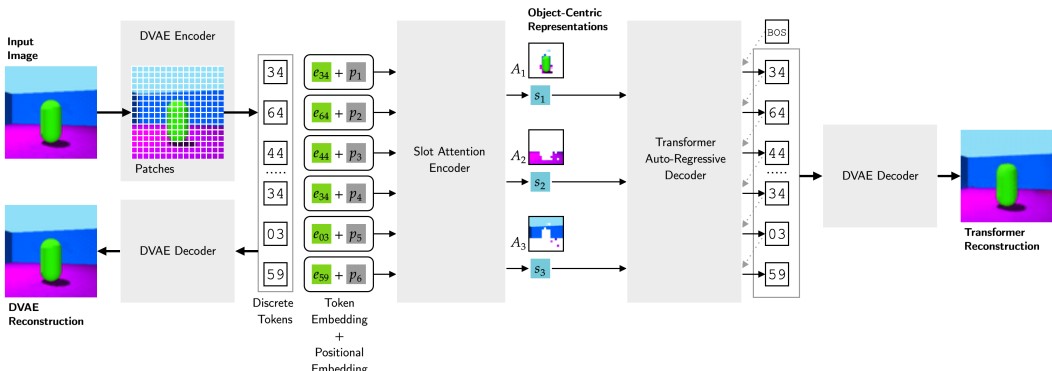

**Figure 8: Architecture of SLATE.** Our model receives an input image and we split it into patches. We encode each patch as a discrete token using a feedforward network (shown as the DVAE encoder). A DVAE decoder is trained to reconstruct each patch from its respective token using a simple MSE loss. We shall call this the DVAE reconstruction and this provides one of the two reconstruction pathways that we have in our architecture. For the second reconstruction pathway, the tokens obtained from the DVAE encoder are mapped to embeddings. To these embeddings, we add learned positional embedding to add the position information to the embedding of each token or the respective image patch. These resulting set of embeddings are provided as input to Slot Attention to discover $N$ slots that summarize the contents of the discrete tokens and thereby summarizing the input image. The slots are then provided to a transformer decoder which is trained to decode and reconstruct the discrete tokens of the input image using a cross-entropy loss. Given the slots, the tokens can be generated by the transformer auto-regressively and then be decoded via the DVAE decoder to produce a reconstruction of the input image. This forms the slot-based reconstruction pathway and, unless specified, the term *reconstruction* would refer to this slot-based pathway.

## A.2  DVAE

**Motivation.** Our motivation for using DVAE is to quantize each patch (4x4 pixels in size) in the image as a discrete token and to ensure that the discrete token can accurately reconstruct the patch via the DVAE decoder. We decided to use DVAE (Ramesh et al., 2021) over VQ-VAE (van den Oord et al., 2017) because DVAE is simpler than VQ-VAE and requires only a simple reconstruction loss to train while VQ-VAE requires additional loss terms. Furthemore, because DALL-E also uses DVAE, it becomes a more natural choice to adopt.

**Omitting the KL Term.** In training the DVAE, we only use the reconstruction loss i.e. a simple MSE reconstruction objective rather than additionally using a KL term to impose a prior on the inferred tokens. This choice is made because we found the training to be more stable when we used smaller $\beta$ coefficients (such as 0.04 or smaller) for the KL term including setting $\beta$ to 0. This suggested that the KL term did not play a useful role in our model and hence setting the $\beta$ coefficient to 0 is a more simpler choice to adopt. This is equivalent to removing the KL term.

## A.3  OBJECT DISCOVERY IN TEXTURED SCENES

In Section 3, we described the encoder for inferring slots $\mathbf{s}_{1:N}$ from a given image $\mathbf{x}$ as follows:

$$\mathbf{o}_i = f_\phi(\mathbf{x}_i)$$
$$\mathbf{z}_i \sim \text{Categorical}(\mathbf{o}_i)$$
$$\mathbf{u}_i = \text{Dictionary}_\phi(\mathbf{z}_i) + \mathbf{p}_{\phi,i}$$
$$\mathbf{s}_{1:N}, A_{1:N} = \text{SlotAttention}_\phi(\mathbf{u}_{1:T}) \ .$$

For simple scenes, it may be sufficient to perform slot attention in this way on the DVAE code embeddings $\mathbf{u}_{1:T}$ as the small receptive field of each code $\mathbf{u}_i$ is enough. However, when dealing with complex and textured images, it is useful to have a CNN that allows pixels belonging to a

much larger receptive field to interact with each other. Thus, for textured datasets namely Textured-MNIST, CelebA and CLEVRTex, we add a CNN to the SLATE encoder as follows:

$$\mathbf{o}_i = f_\phi(\mathbf{x}_i)$$
$$\mathbf{z}_i \sim \text{Categorical}(\mathbf{o}_i)$$
$$\mathbf{e}_i = \text{Dictionary}_\phi(\mathbf{z}_i)$$
$$\mathbf{u}_{1:T}^{\text{interact}} = \text{Flatten}(\text{CNN}(\text{Reshape}(\mathbf{e}_{1:T}))) + \mathbf{p}_{\phi,1:T}$$
$$\mathbf{s}_{1:N}, A_{1:N} = \text{SlotAttention}_\phi(\mathbf{u}_{1:T}^{\text{interact}})$$

Here, Reshape($\cdot$) reshapes the DVAE embedding sequence $\mathbf{e}_{1:T}$ to a 2D feature map. This feature map is fed to a CNN which outputs a feature map of the same size. The output feature map is then flattened and positional embeddings are added to it to obtain a sequence of embeddings $\mathbf{u}_{1:T}^{\text{interact}}$. These embeddings are then provided to slot attention for inferring the slots. We describe the CNN architecture that we use in Table 3. For auto-regressive decoding, the inputs to the transformer are $\mathbf{u}_i = \text{Dictionary}_\phi(\mathbf{z}_i) + \mathbf{p}_{\phi,i}$ as before as described in Section 3.

| Layer | Stride | Channels | Activation |
|---|---|---|---|
| Conv $3 \times 3$ | 1x1 | 192 | ReLU |
| Conv $3 \times 3$ | 1x1 | 192 | ReLU |
| Conv $3 \times 3$ | 1x1 | 192 | ReLU |
| Conv $3 \times 3$ | 1x1 | 192 | ReLU |
| Conv $3 \times 3$ | 1x1 | 192 | ReLU |

**Table 3: CNN Architecture for Textured-MNIST, CelebA and CLEVRTex.** The number of channels, i.e. 192, is the same as the size of the DVAE embedding that the CNN receives as input.

### A.4 MULTI-HEADED SLOT ATTENTION

When learning the slots using standard Slot Attention encoder (Locatello et al., 2020), we found that slots can suffer in expressiveness when representing objects with complex shapes and textures. This is because the slots collect information from the input cells via dot-product attention in which the attended input values are pooled using a simple weighted sum. As such, this pooling method can be too weak for representing complex objects. To address this, we propose an extension of Slot Attention called *Multi-Headed Slot Attention* in which each slot attends to the input cells via multiple heads. Thus each slot can attend to different parts of the same object. When the slot state is computed, the attended values of different heads are concatenated which allows these partial object features to interact more flexibly and produce a significantly better object representation. Below, we provide the details of the implementation and an ablation experiment to show the benefits of multiple heads.

In Slot-Attention (Locatello et al., 2020), each slot undergoes several rounds of interactions with the input image in order to collect the contents for the slot representation. In each iteration, this interaction of a slot with the input image is performed using dot-product attention with slot as the query. The value returned by this attention is a simple weighted sum of the input cells being attended by the slot. This weighted sum as a pooling method and method of interaction between the attended cells can be insufficient for properly representing the attended cells. This is especially the case when the object has a complex shape and texture. To address this, we propose a multi-headed extension of slot attention. In this, we replace the dot-product attention with a multi-headed dot-product attention. By doing this, each slot attends to the different parts of the same object simultaneously and the result of the attention is concatenated. As this concatenated result passes through the RNN layers, these features for different parts of the same object can interact flexibly and result in a more rich encoding of the attended object. Recall that in standard slot attention, because there is only one attention head, different parts of the same object can only interact via the weighted sum which serves as a weak form of interaction as our results below shall indicate.

**Implementation.** We implement iterations of the Multi-headed Slot Attention as follows. For a given iteration, we first project the slots $\mathbf{S} = \mathbf{s}_{1:N}$ from the previous iteration into query vectors for

| Model | MSE |
|---|---|
| **Ours** (`num_slot_heads=1, num_slots=8`) | 391.15 |
| **Ours** (`num_slot_heads=1, num_slots=15`) | 371.05 |
| **Ours** (`num_slot_heads=4, num_slots=8`) | **136.29** |

**Table 4:** Effect of number of slots and number of slot heads on the reconstruction quality in Bitmoji dataset in SLATE. We compare the benefits of increasing the number of heads in our slot attention module and the effect of increasing the number of slots. We note that simply increasing the number of slots (from 8 to 15) does not improve the reconstruction quality significantly. However, increasing the number of heads (from 1 to 4) while keeping the number of slots same significantly improves the reconstruction quality.

| Model | MSE |
|---|---|
| **SA** (`num_slot_heads=1, num_slots=8`) | 394.5 |
| **SA** (`num_slot_heads=4, num_slots=8`) | 410.1 |

**Table 5:** Effect of number of slot heads on the reconstruction quality in Bitmoji dataset using Slot Attention (SA) with mixture-decoder. We note that with mixture decoder, having multiple heads does not benefit reconstruction as the performance bottleneck is the weak object component decoder based on Spatial Broadcast decoder (Watters et al., 2019)

each head. Similarly, we project the input cells $\mathbf{U} = \mathbf{u}_{1:T}$ into key and value vectors for each head.

$$\mathbf{Q}_m = \mathbf{W}_m^Q \mathbf{S},$$
$$\mathbf{K}_m = \mathbf{W}_m^K \mathbf{U},$$
$$\mathbf{V}_m = \mathbf{W}_m^V \mathbf{U}.$$

Next, we compute unnormalized attention proportions for each head $m$ as by taking a dot-product as follows:

$$\frac{\mathbf{Q}_m \mathbf{K}_m^T}{\sqrt{D_K}}.$$

where $D_K$ is the size of the key vectors. The attention map over the input cells $\mathbf{A}_{n,m}$ for slot $n$ and head $m$ is obtained by taking a softmax over both the $N$ slots and the $M$ heads.

$$[\mathbf{A}_{1:N,1}; \ldots; \mathbf{A}_{1:N,M}] = \text{softmax}\left(\left[\frac{\mathbf{Q}_1 \mathbf{K}_1^T}{\sqrt{D_K}}; \ldots; \frac{\mathbf{Q}_M \mathbf{K}_M^T}{\sqrt{D_K}}\right], \texttt{dim} = \text{`slots' and `heads'}\right).$$

We then obtain the attention result for a given slot $n$ as follows.

$$\texttt{updates}_n = [\mathbf{A}_{n,1}\mathbf{V}_1; \ldots; \mathbf{A}_{n,M}\mathbf{V}_M]\mathbf{W}^O \tag{1}$$

where and $\mathbf{W}^O$ is an output projection matrix. With this update vector, the slot state is updated as follows as done in the standard Slot Attention (Locatello et al., 2020).

$$\mathbf{s}_n \leftarrow \text{GRU}(\texttt{updates}_n, \mathbf{s}_n),$$
$$\mathbf{s}_n \leftarrow \mathbf{s}_n + \text{MLP}(\text{LayerNorm}(\mathbf{s}_n)).$$

This results in the update slot that is given to the next iteration.

**Results.** We apply Multi-headed Slot Attention in the Bitmoji dataset for our model. We found significant improvement in the MSE when we used 4 heads as compared to the standard Slot Attention (equivalent to 1 head). This is reported in Table 4.

We also noted that when using mixture decoder, using multi-headed slot attention does not help because the performance bottleneck is the Spatial Broadcast decoder (Watters et al., 2019) of the mixture decoder. This comparison is shown in Table 5.

**Limitations.** While multi-head attention helps information in the slots, in datasets with simple objects, having multiple heads can harm disentanglement of objects into different slots. This can occur because as each slots becomes more expressive, the network may be incentivised to collect

| Model | FID |
|---|---|
| Our Model (VQ Input + Transformer Decoder) | 36.75 |
| Ablation (CNN Input + Transformer Decoder) | 32.07 |
| Ablation (VQ Input + Mixture Decoder) | 174.83 |
| Slot Attention (CNN Input + Mixture Decoder) | 44.14 |

**Table 6: Ablation of Model Architecture in 3D Shapes.** We ablate our model by replacing the discrete token input (VQ) with a CNN feature map as originally used by Locatello et al. (2020). We also ablate our model by replacing the Transformer decoder with the mixture decoder. These results suggest that the main driver of generation quality is the use of Transformer decoder as both models having the transformer decoder outperform both the models having the mixture decoder in terms of the FID score.

information about multiple objects into the same slot. Therefore for our experiments on 3D Shapes, CLEVR and Shapestacks datasets, we used regular slot attention module with 1 head. Because Bitmoji can have complex hair and face shapes, we used 4-headed slot attention module for this dataset in our model.

### A.5 ABLATION OF ARCHITECTURE

To better justify our design, we perform two additional experiments that we describe here. In terms of architectural components, our model can be seen as VQ-Input + Slot Attention + Transformer Decoder while our baseline (Locatello et al., 2020) can be seen as CNN Input + Slot Attention + Mixture Decoder. Hence, the components that differ between our model and the baseline are *i)* providing VQ-Input and *ii)* using a Transformer decoder. We analyze the effect of these individual components by performing two ablations. In the first ablation, we take our model and replace the VQ-Input with CNN Input. We refer to this ablation as *CNN Input + Transformer Decoder*. In the second ablation, we take our model and replace the Transformer decoder with the mixture decoder. We refer to this ablation as *VQ-Input + Mixture Decoder*. We report the results in Table 6. We also visualize the object attention maps of the ablation models in Figure 33.

## B ADDITIONAL RELATED WORK

**Compositional Generation with Latent Variable Models and Text.** Early approaches focused on disentangling the independent factors of the observation generating process using $\beta$-VAE (Higgins et al., 2017) or ensuring that independently sampling each latent factor should produce images indistinguishable from the training distribution (Kim & Mnih, 2018; Kumar et al., 2017; Chen et al., 2018). However unlike ours, these approaches providing a monolithic single vector representation can suffer in multi-object scenes due to *superposition catastrophe* (Greff et al., 2020a; Jiang & Ahn, 2020). Another line of approaches learn to map text to images to compose novel scenes (Ramesh et al., 2021; Reed et al., 2016b; Li et al., 2019; Higgins et al., 2018) or conditional image generation (Sohn et al., 2015; Yan et al., 2016; Johnson et al., 2018; Ashual & Wolf, 2019; Bar et al., 2021; Herzig et al., 2020; 2019; Du et al., 2020). However such approaches rely on supervision.

## C HYPERPARAMETERS AND COMPUTATIONAL REQUIREMENTS

We report the hyperparameters and the computational resources required for training our model in Table 8.

**Dropout.** We found it beneficial to perform dropout during training of our model. We apply attention dropout of 0.1 in the transformer decoder. We also apply the same dropout of 0.1 after positional embedding is added to the patch token embedding i.e.

$$\mathbf{u}_i \leftarrow \text{Dropout}(\text{Dictionary}_\phi(\mathbf{z}_i) + \mathbf{p}_{\phi,i}, \ \texttt{dropout=0.1}).$$

**Learning Rate Schedules.** For training the parameters of DVAE, we noted that a constant learning rate of 3e-4 produced good discretization of patches and low reconstruction error of the patches decoded from the DVAE code. Smaller learning rates lead to poorer use of the dictionary and poorer reconstruction of patches.

| Dataset | | 3D Shapes | CLEVR | Shapestacks | Bitmoji |
|---|---|---|---|---|---|
| Training Images | | 400K | 200K | 230K | 100K |
| Batch Size | | 50 | 50 | 50 | 50 |
| LR Warmup Steps | | 30000 | 30000 | 30000 | 30000 |
| Peak LR | | 0.0001 | 0.0003 | 0.0003 | 0.0001 |
| Dropout | | 0.1 | 0.1 | 0.1 | 0.1 |
| DVAE | Vocabulary Size | 1024 | 4096 | 4096 | 4096 |
| | Temp. Cooldown | 1.0 to 0.1 | 1.0 to 0.1 | 1.0 to 0.1 | 1.0 to 0.1 |
| | Temp. Cooldown Steps | 30000 | 30000 | 30000 | 30000 |
| | LR (no warmup) | 0.0003 | 0.0003 | 0.0003 | 0.0003 |
| Image Size | | 64 | 128 | 96 | 128 |
| Image Tokens | | 256 | 1024 | 576 | 1024 |
| Transformer Decoder Specifications | Layers | 4 | 8 | 8 | 8 |
| | Heads | 4 | 8 | 8 | 8 |
| | Hidden Dim. | 192 | 192 | 192 | 192 |
| Slot Attention Specifications | Slots | 3 | 12 | 12 | 8 |
| | Iterations | 3 | 7 | 7 | 3 |
| | Slot Heads | 1 | 1 | 1 | 4 |
| | Slot Dim. | 192 | 192 | 192 | 192 |
| Training Cost | GPU Usage | 8GB | 64GB | 14GB | 64GB |
| | Days | 11 hours | 5 Days | 5 Days | 3.5 Days |

**Table 7:** Hyperparameters used for our model and computation requirements for 3D Shapes, CLEVR-Mirror, Shapestacks and Bitmoji.

| Dataset | | TexMNIST | CelebA | CLEVRTex |
|---|---|---|---|---|
| Training Images | | 175K | 140K | 35K |
| Batch Size | | 50 | 50 | 50 |
| LR Warmup Steps | | 30000 | 30000 | 30000 |
| Peak LR | | 0.0003 | 0.0003 | 0.0003 |
| Dropout | | 0.1 | 0.1 | 0.1 |
| DVAE | Vocabulary Size | 4096 | 4096 | 4096 |
| | Temp. Cooldown | 1.0 to 0.1 | 1.0 to 0.1 | 1.0 to 0.1 |
| | Temp. Cooldown Steps | 30000 | 30000 | 30000 |
| | LR (no warmup) | 0.0003 | 0.0003 | 0.0003 |
| Image Size | | 64 | 128 | 128 |
| Image Tokens | | 256 | 1024 | 1024 |
| Transformer Decoder Specifications | Layers | 4 | 8 | 8 |
| | Heads | 4 | 8 | 8 |
| | Hidden Dim. | 192 | 192 | 192 |
| Slot Attention Specifications | Slots | 4 | 4 | 12 |
| | Iterations | 3 | 3 | 3 |
| | Slot Heads | 1 | 1 | 1 |
| | Slot Dim. | 192 | 192 | 192 |
| Training Cost | GPU Usage | 8GB | 64GB | 64GB |
| | Time | 16 hours | 3 Days | 3.5 Days |

**Table 8:** Hyperparameters used for our model and computation requirements for Textured-MNIST, CelebA and CLEVRTex.

For the weights of slot attention encoder, the transformer and the learned embeddings, we found a learning rate warm-up schedule helpful. For this warm-up, we increase the learning rate linearly from 0.0 to the peak learning rate in the first 30K training steps. After this, at the end of every epoch, we monitor the loss on the validation set. If the validation loss does not decrease for 8 consecutive epochs, we reduce the learning rate by a factor of 1/2.

# D    ADDITIONAL QUALITATIVE RESULTS

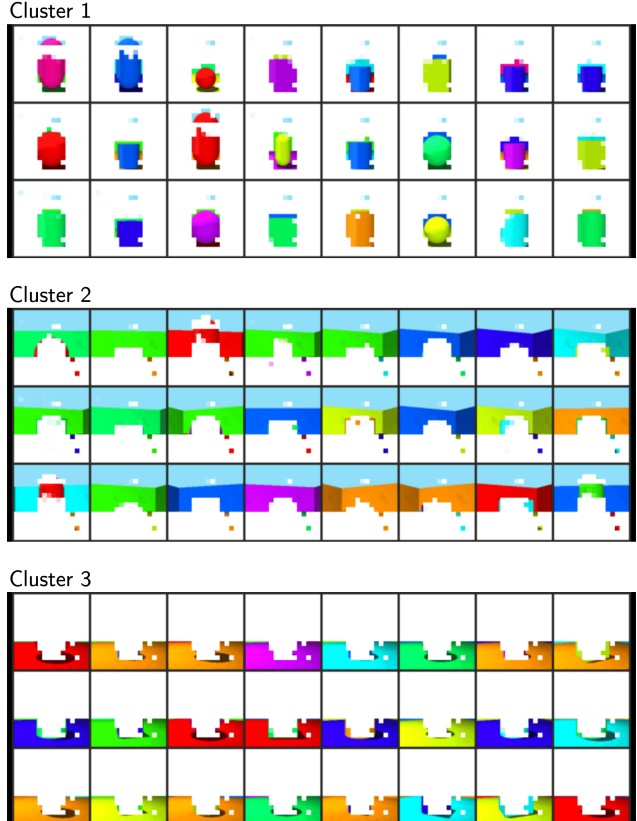

**Figure 9:** Clusters in the concept library obtained by applying $K$-means on slots obtained from the 3D Shapes dataset. We note that our slot representation space models the semantics as the objects of the same class tend to have more similar representation and automatically form a cluster when $K$-means is applied.

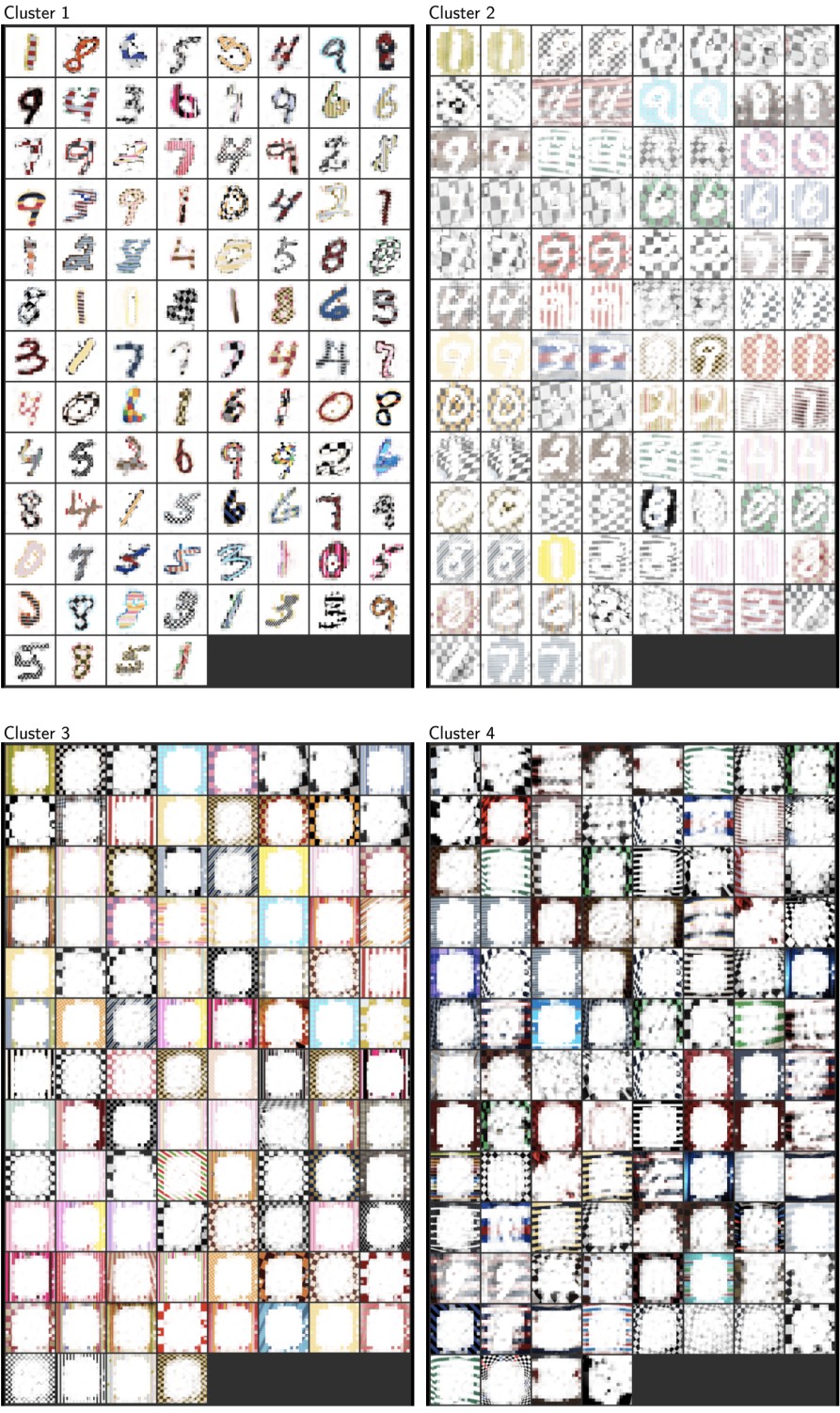

**Figure 10:** Clusters in the concept library obtained by applying $K$-means on slots obtained from the Textured MNIST dataset.

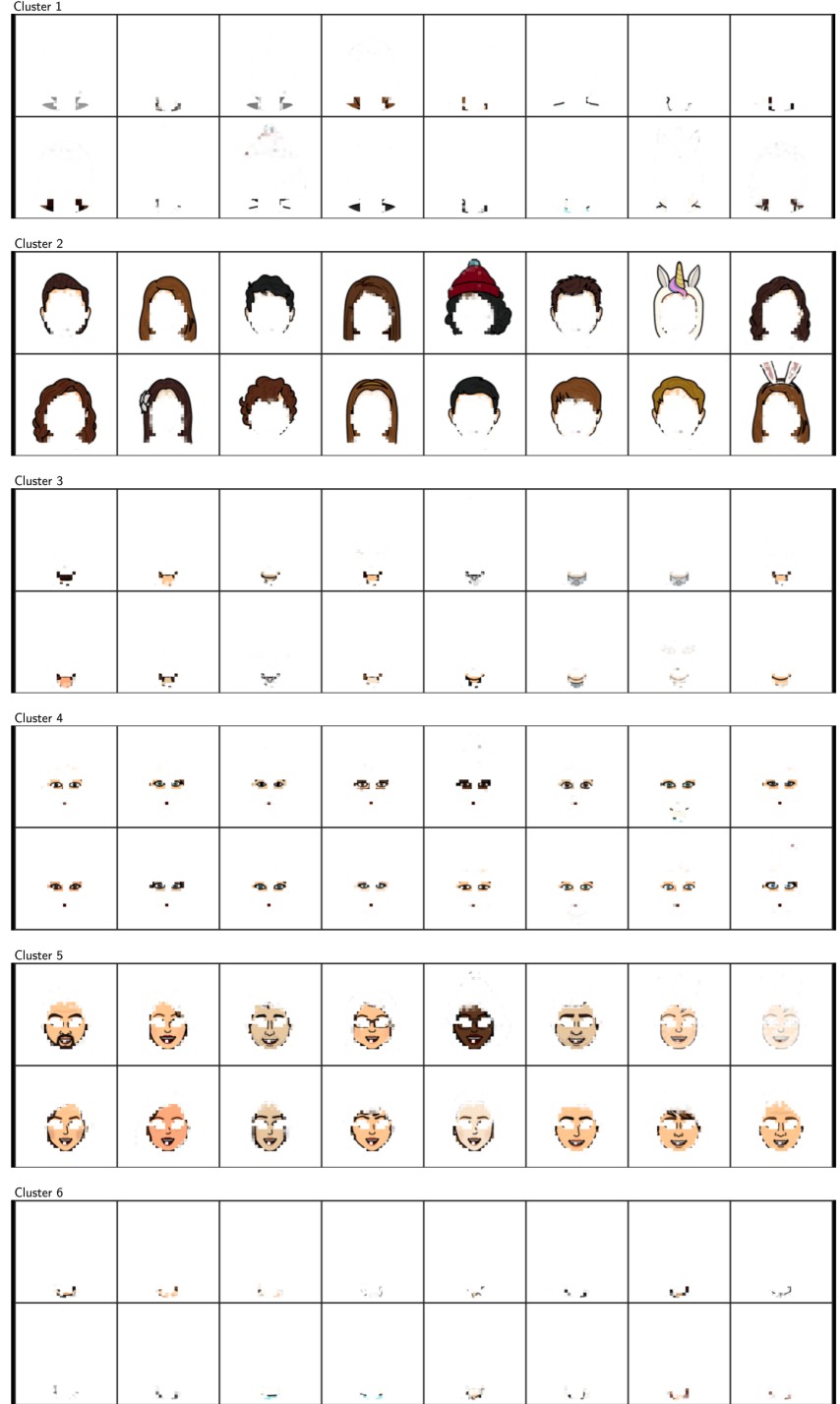

**Figure 11:** Clusters in the concept library obtained by applying $K$-means on slots obtained from the Bitmoji dataset. We note that our slot representation space models the semantics as the objects of the same class tend to have more similar representation and automatically form a cluster.

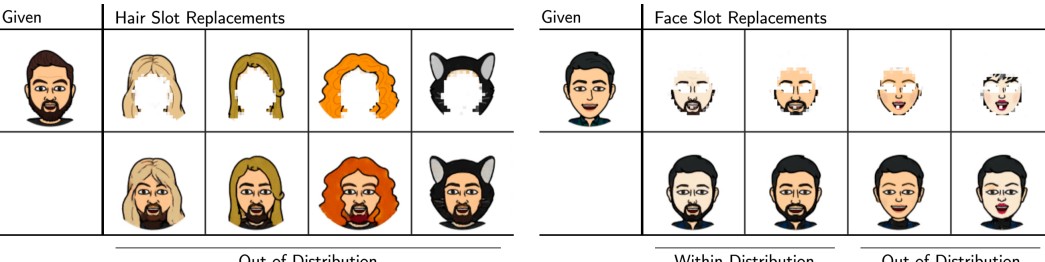

**Figure 12: Object Replacement and Out-of-Distribution Compositions in Bitmoji.** We show that in our model, it is possible to edit the image by taking a specific slot and replacing it with an arbitrary slot drawn from the concept library for the same concept. We also show OOD compositions by replacing the slot with those from the opposite gender.

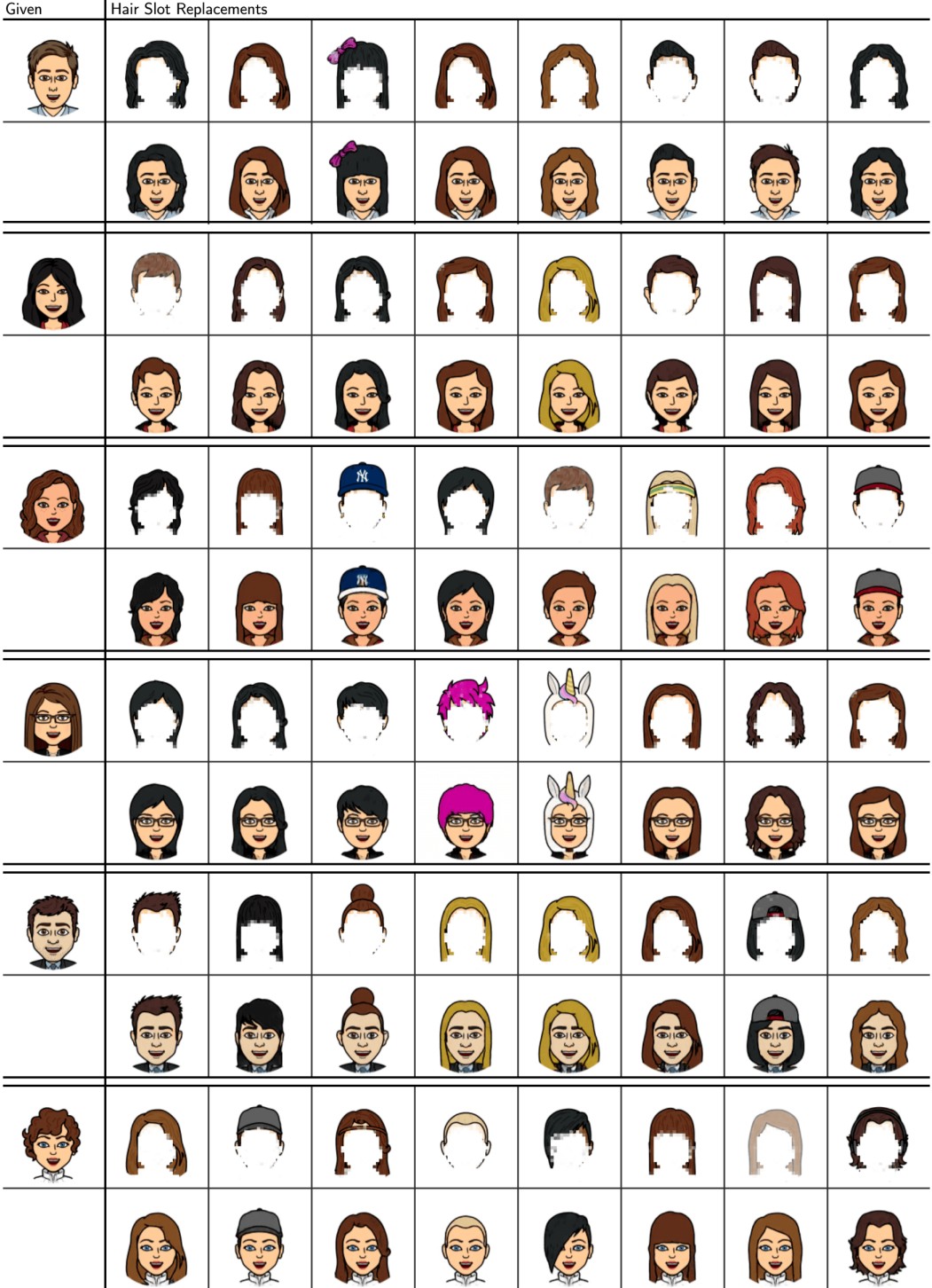

**Figure 13:** Qualitative Results for Hair Slot Replacements in Bitmoji dataset. For each image, the first row shows the hair slot that will be used as a replacement and the second row shows the generation from our model after the hair slot is replaced. This also provides examples of OOD compositions when the given image has the opposite gender from that of the source image from which the replacement slot is taken.

Slot Prompt                                                                 Composition

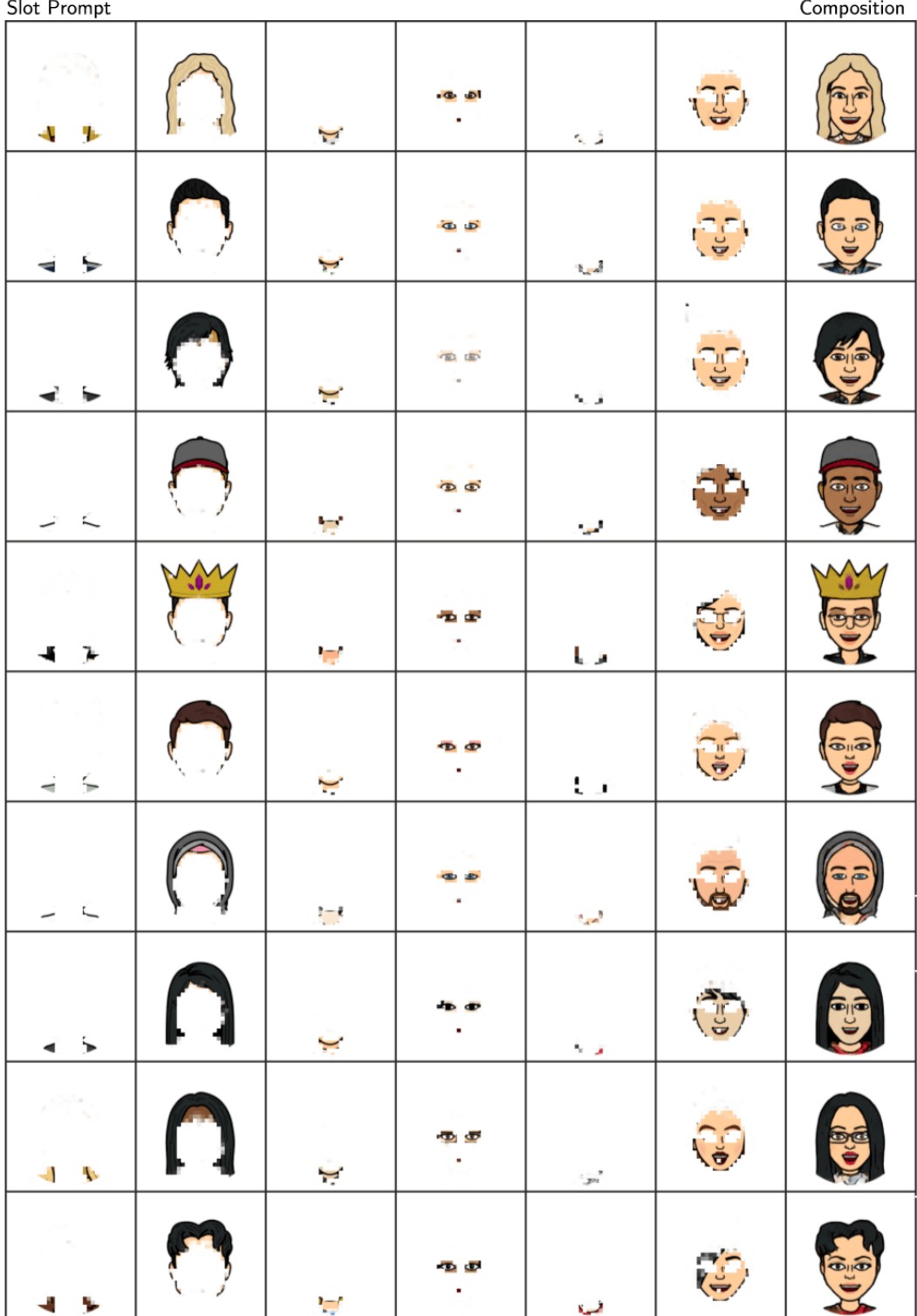

**Figure 14:** Qualitative Results for Compositional Generation in Bitmoji dataset using our model.

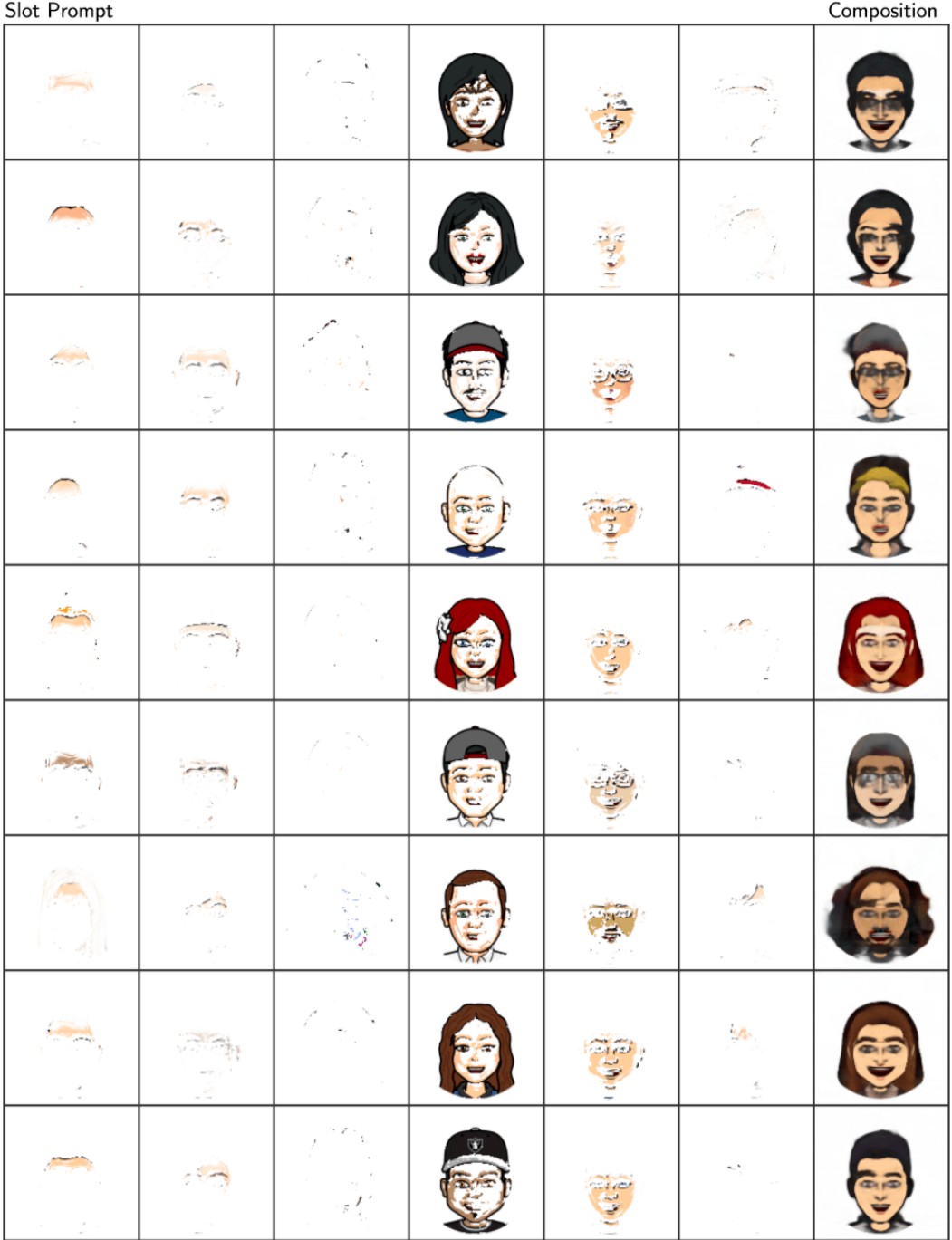

**Figure 15:** Qualitative Results for Compositional Generation in Bitmoji dataset using Slot Attention.

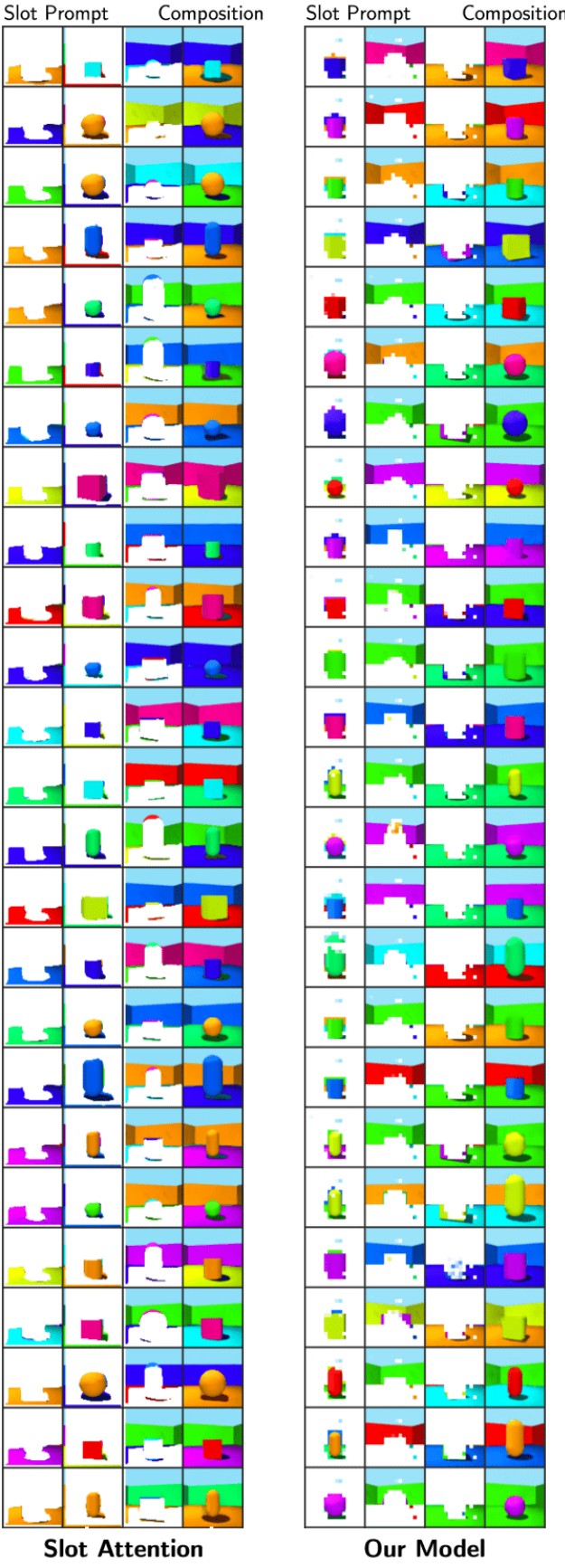

**Figure 16:** Qualitative Comparison of Compositional Generation in 3D Shapes.

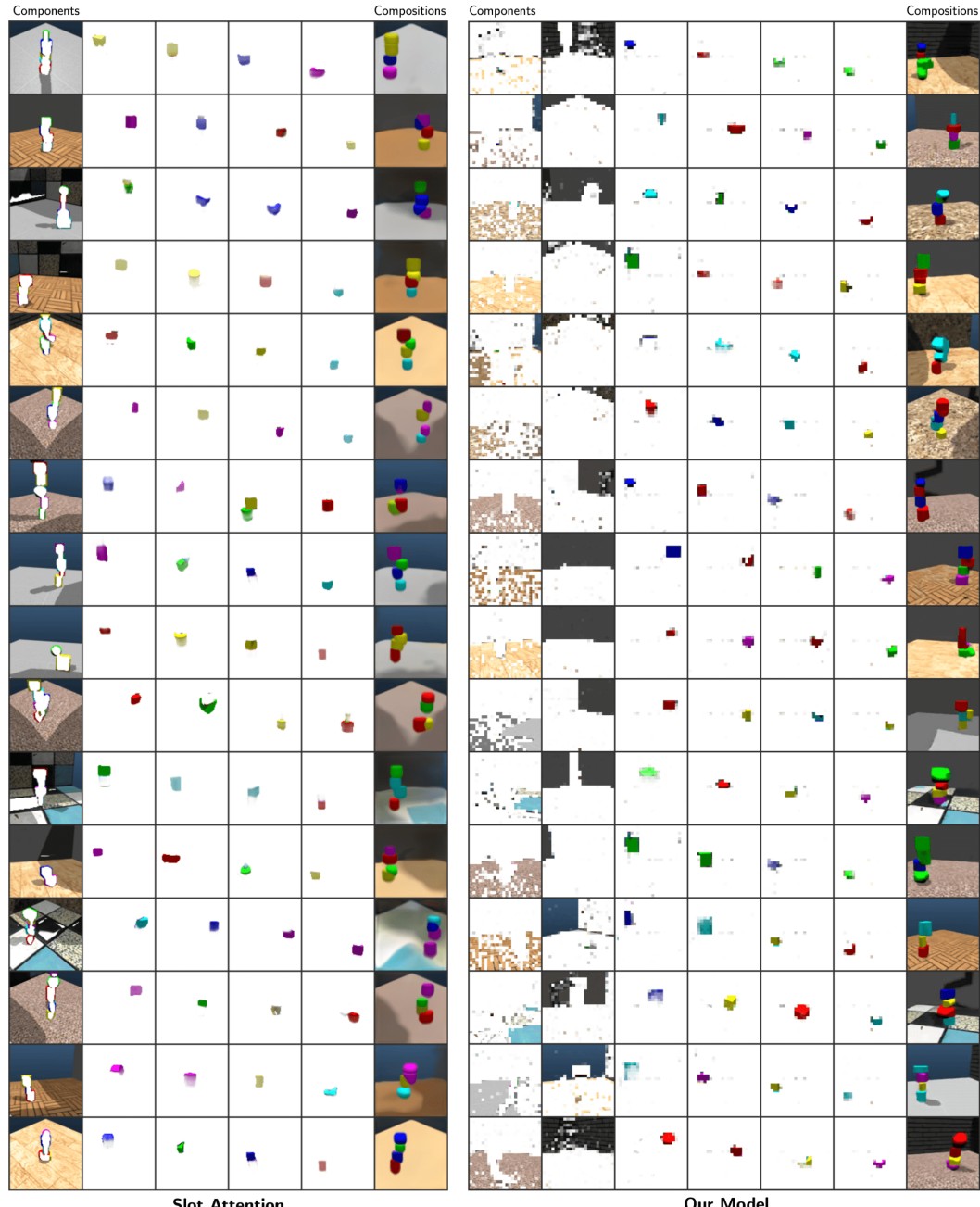

**Figure 17:** Qualitative Comparison of Compositional Generation in Shapestacks.

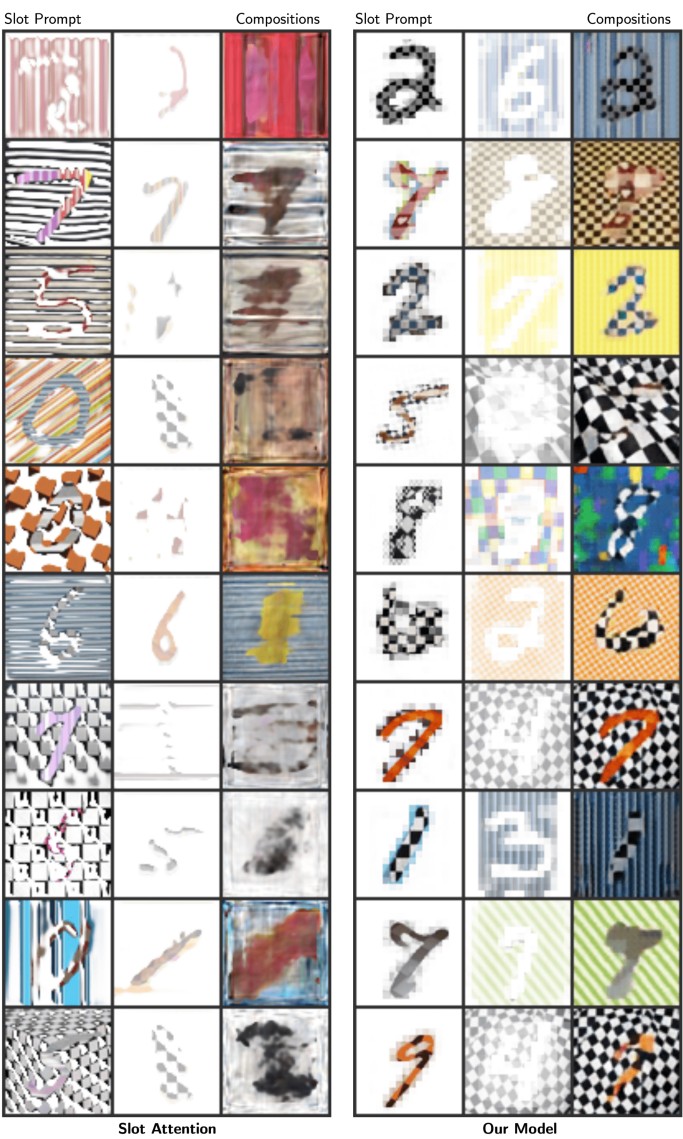

**Figure 18:** Qualitative Comparison of Compositional Generation in TexturedMNIST.

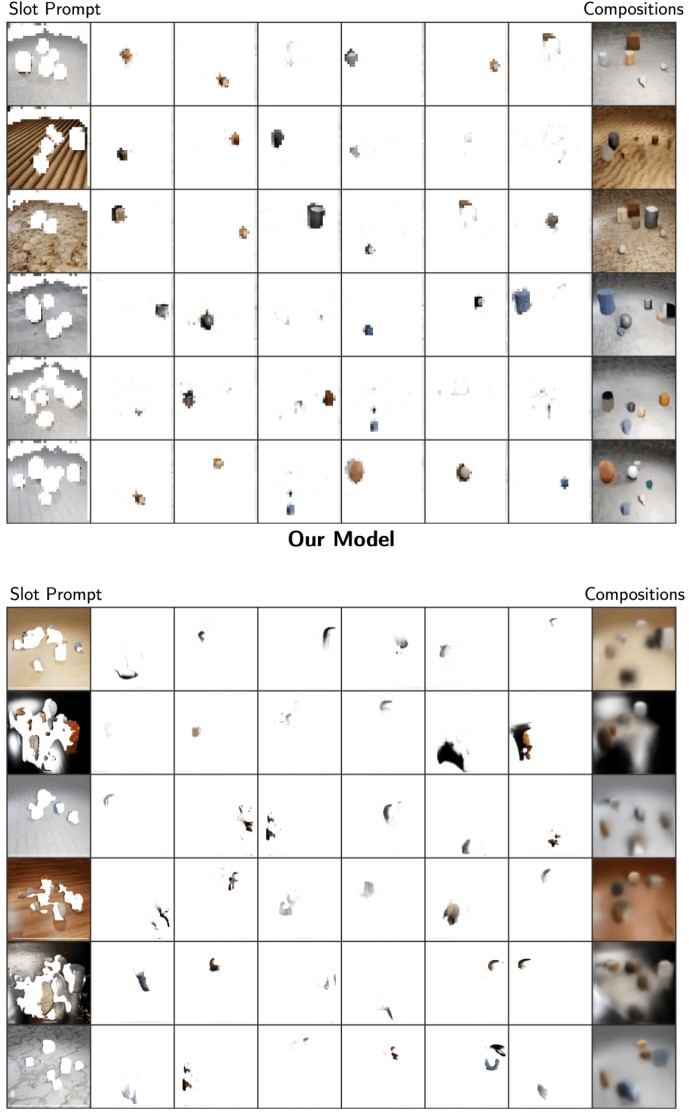

**Figure 19:** Qualitative comparison of compositional generation in CLEVRTex.

Slot Prompt

Composition

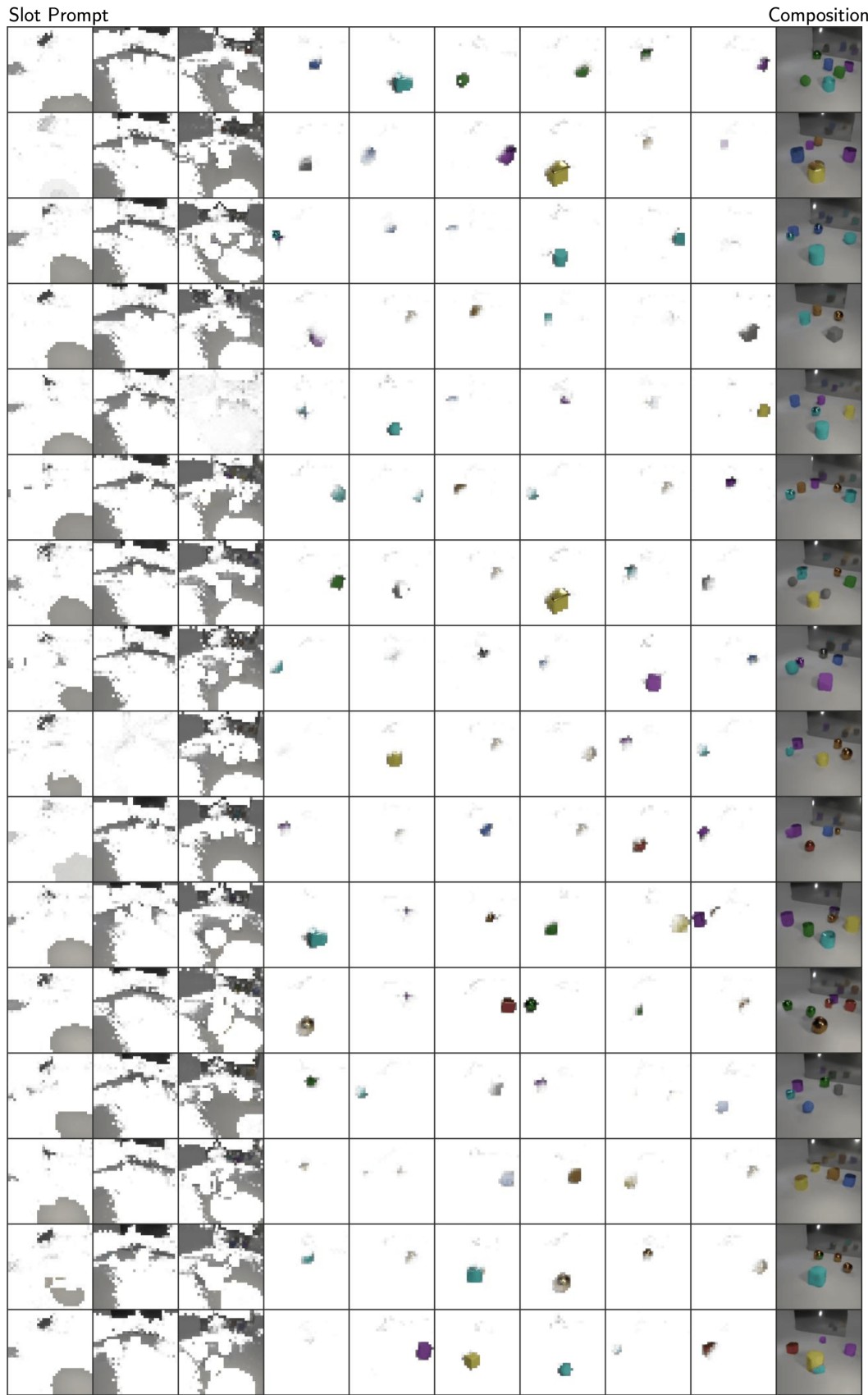

**Figure 20:** Qualitative Samples of Compositional Generation in CLEVR from our model.

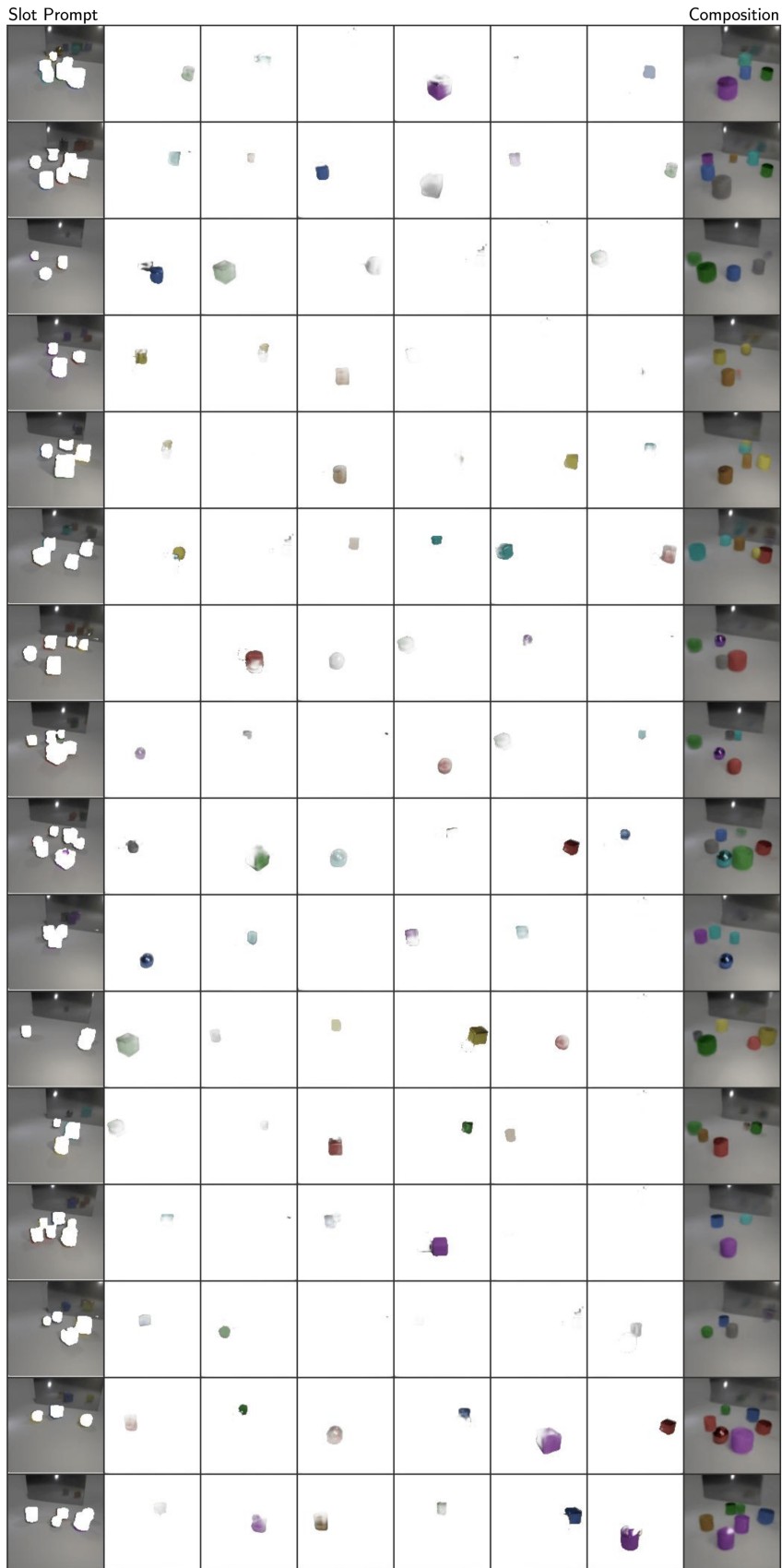

**Figure 21:** Qualitative Samples of Compositional Generation in CLEVR from Slot Attention with pixel-mixture decoder.

Slot Prompt                                                                                          Compositions

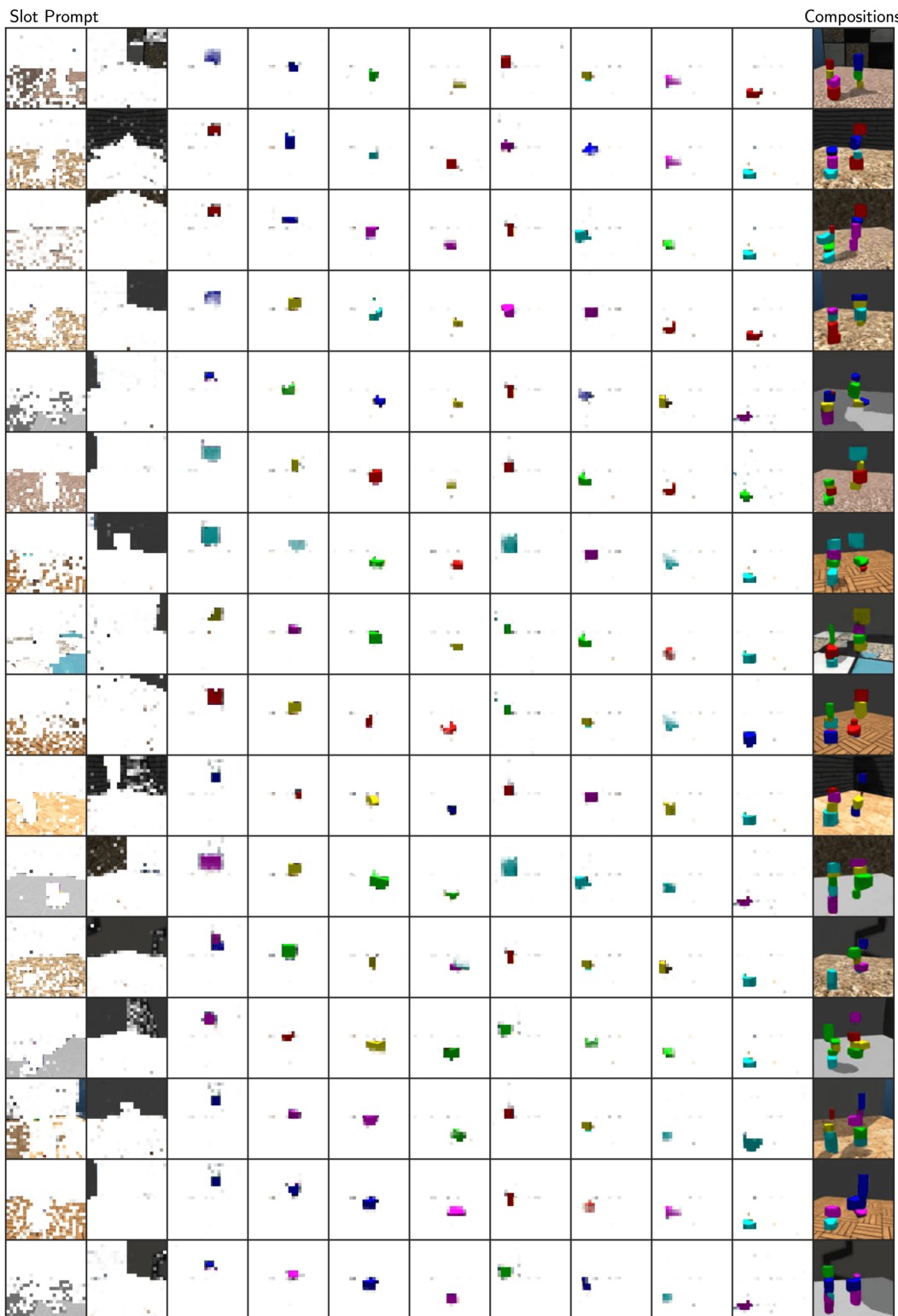

**Figure 22:** Qualitative Results for Out-of-Distribution Compositional Generation in Shapestacks from our model.

Slot Prompt
Compositions

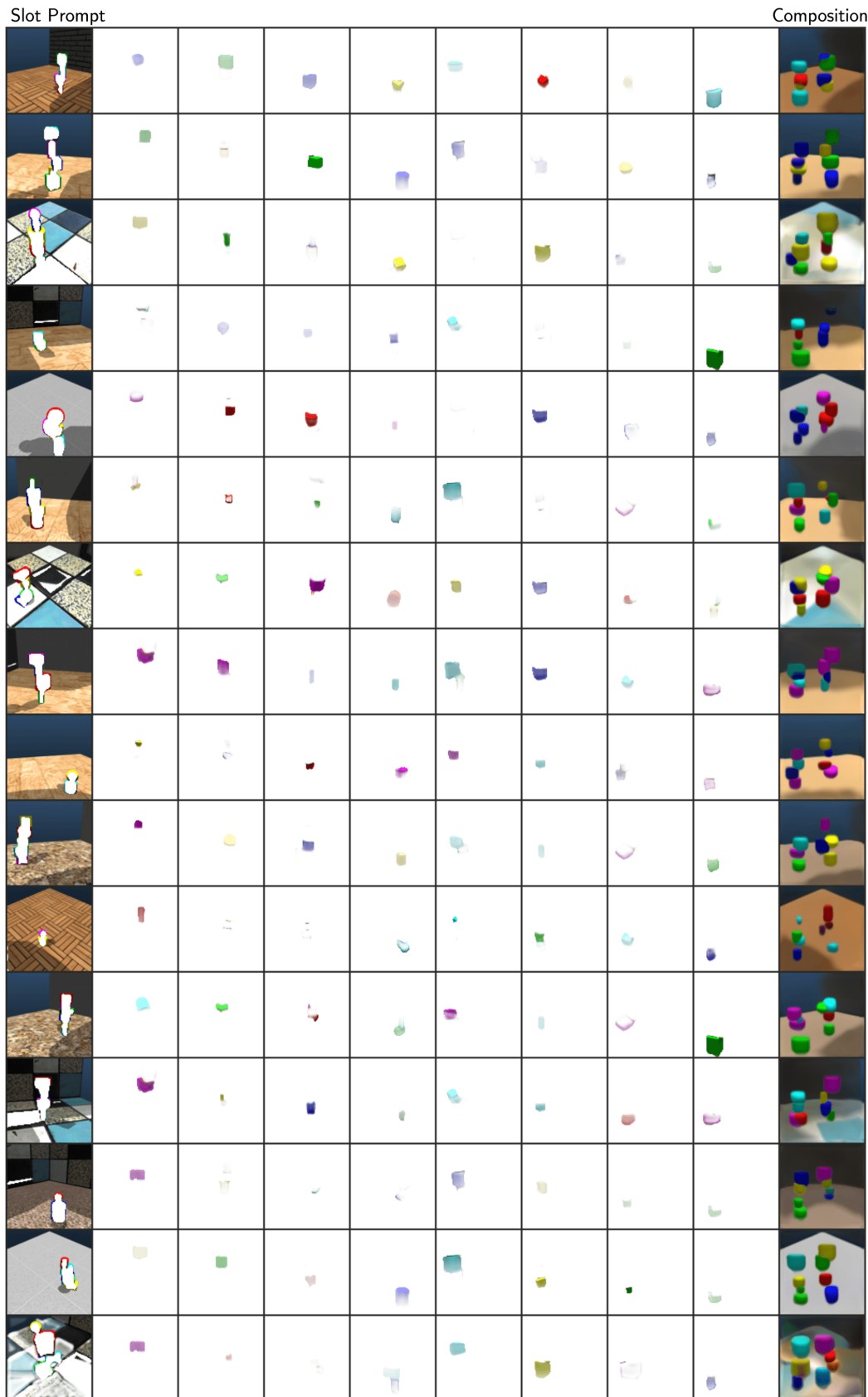

**Figure 23:** Qualitative Results for Slot Attention for Out-of-Distribution Compositional Generation in Shapestacks.

# E    ADDITIONAL DISCUSSION

## E.1    STRONG DECODER IN MIXTURE LIKELIHOOD MODELS MAKE SLOTS CAPTURE MULTIPLE OBJECTS

One of the attractive properties of our model is that our transformer decoder produces images with significantly better detail than Slot Attention. Slot Attention fails to render such details largely because of Spatial Broadcast Decoder that is used to render each object component. Spatial Broadcast Decoder decodes images by mapping a pixel coordinate and the provided slot vector to RGB pixel values and a mask using an MLP. This MLP prefers to learn simple mappings from pixel coordinate to RGB values. At the start of the training, this function tends to be a constant or a linear function of pixel coordinate and only after more training does this function achieve a more complex input-output behavior.

Hence, when a Spatial Broadcast Decoder is used, the decoder of each object component is biased in early training to prefer modeling segments without much RGB variation within the object patch. This property is important to guide the training of Slot Attention encoder to learn to disentangle separate objects and it has been exploited in several prior works on unsupervised object-centric representation learning. To understand why this occurs, consider a grayscale image that contains two balls, one colored white with pixel value 1.0 and the other colored grey with pixel value 0.5. Now consider the case when each slot represents one object in the scene. In this case, the decoder for the white ball simply needs to output a value 1.0 for all pixel coordinates and this is a constant function. Similarly, for the grey ball, the decoder needs to output a value 0.5 for all pixel coordinates. In contrast, consider the case where a single slot represents both the white and the grey balls. In this case, the decoder needs to map the pixels coordinates for the white ball to 1.0 and the pixel coordinates for the grey ball to 0.5 within the same function. This function is not longer a constant and is therefore much more complex than the functions that arose when each slot modeled separate objects. Furthermore, as objects might be positioned randomly, this mapping needs to quickly change when the positions of the balls change in the scene depending on information from the slot representation. This makes the mapping even more complex when more than one objects are modeled by the same slot. Hence, the network naturally prefers to model and decode different objects via different slots. As a result, the encoder is incentivized to cluster the pixels of the input image into objects. While this property could be utilized in simple datasets which have objects without much RGB variation and detail within the object, it can fail to render images properly in datasets such as Shapestacks with floor with complex textures or Bitmoji with complex face details and the shape of the hair.

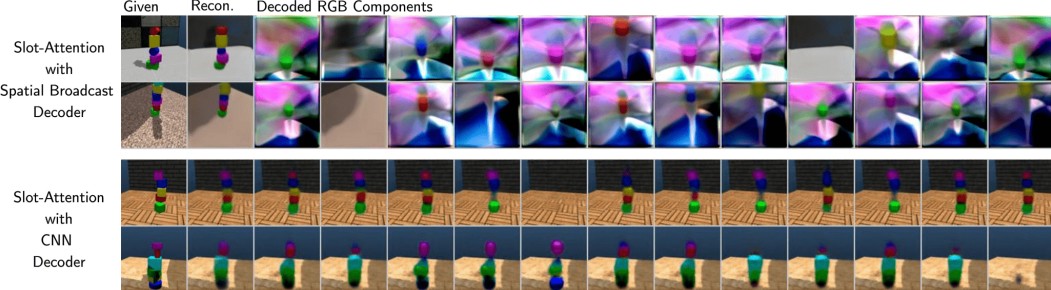

**Figure 24:** Analysis of decoder capacity in Slot Attention when using a mixture-based likelihood model for decoding. We note that when we use a powerful CNN decoder to decode the RGB components and decoding masks for the objects, then a single component tries to model all the objects in the scene instead of each slot representing a single object as in the case of Spatial Broadcast decoder.

While one may argue that this could be fixed simply by replacing the Spatial Broadcast Decoder with a more powerful decoder for the object components. However, because weak decoder was encouraging the disentanglement of objects as argued above, hence by the same argument, when a powerful decoder is used, a single slot no longer has the incentive to model one object. Thus each slot can try to represent and model more than one object in the scene. We tested this empirically by replacing the Spatial Broadcast Decoder of Slot Attention with a powerful CNN decoder. We found

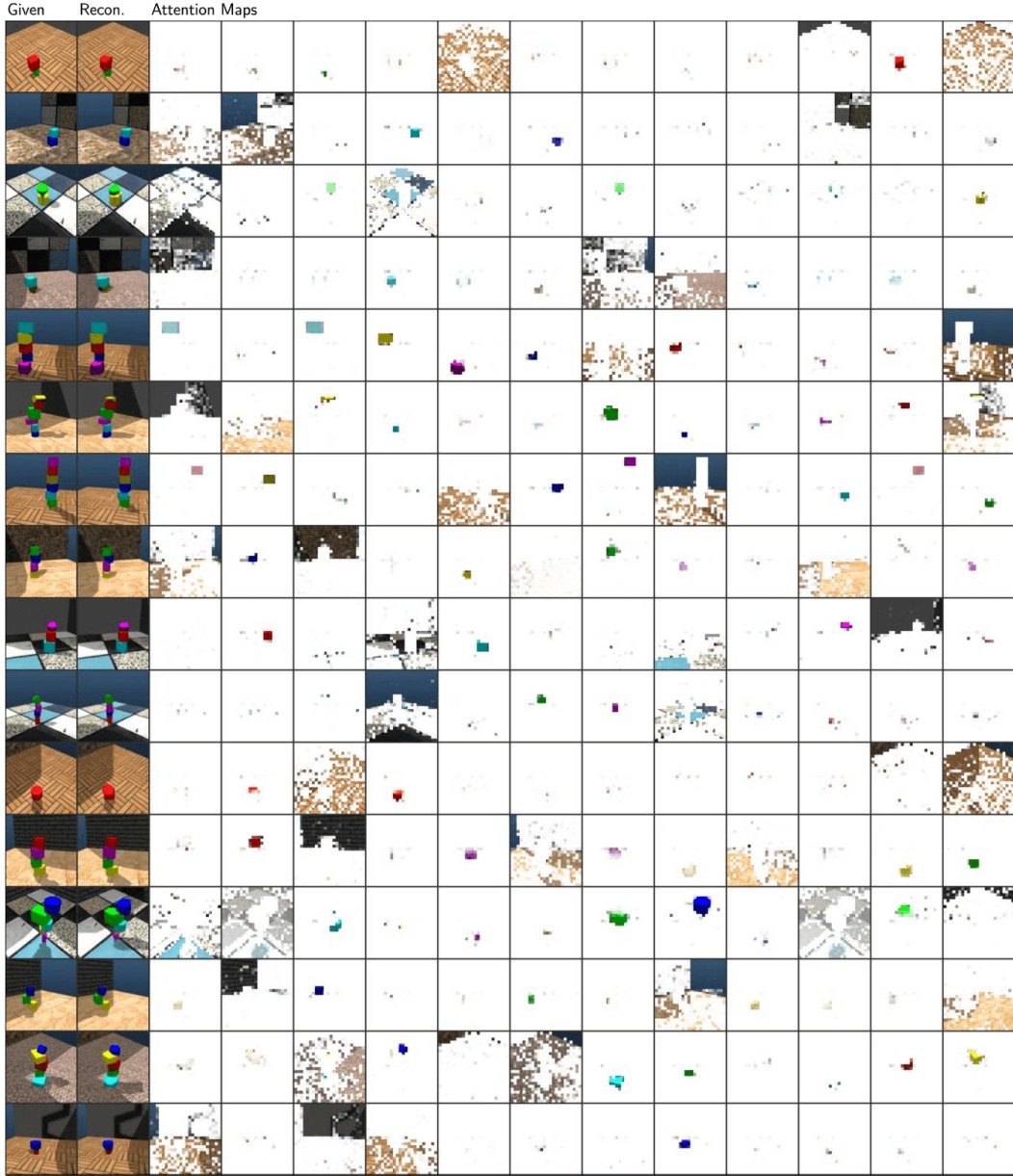

**Figure 25:** Reconstruction and attention maps of slots in SLATE given the input images for Shapestacks. We note that attention maps effectively localize on the individual blocks.

that this causes the slots to capture multiple objects and in some cases a single slot may try to model all the contents of the scene as a single object segment. This is shown in Figure 24.

Hence for unsupervised object discovery, this historically presented a trade-off between decoding capacity and the ability of the model to disentangle the objects. In contrast, our work shows that such a trade-off can be eliminated if transformer is used as the decoder. Our model can not only detect objects without supervision but also render their fine details during decoding and thus resolves the pixel-decoding dilemma mentioned in Section 2.2. We show qualitative samples of reconstructions from our model and the input attention maps of slots in Figures 25, 26, 27, 28, 29, 30, 31, and 32.

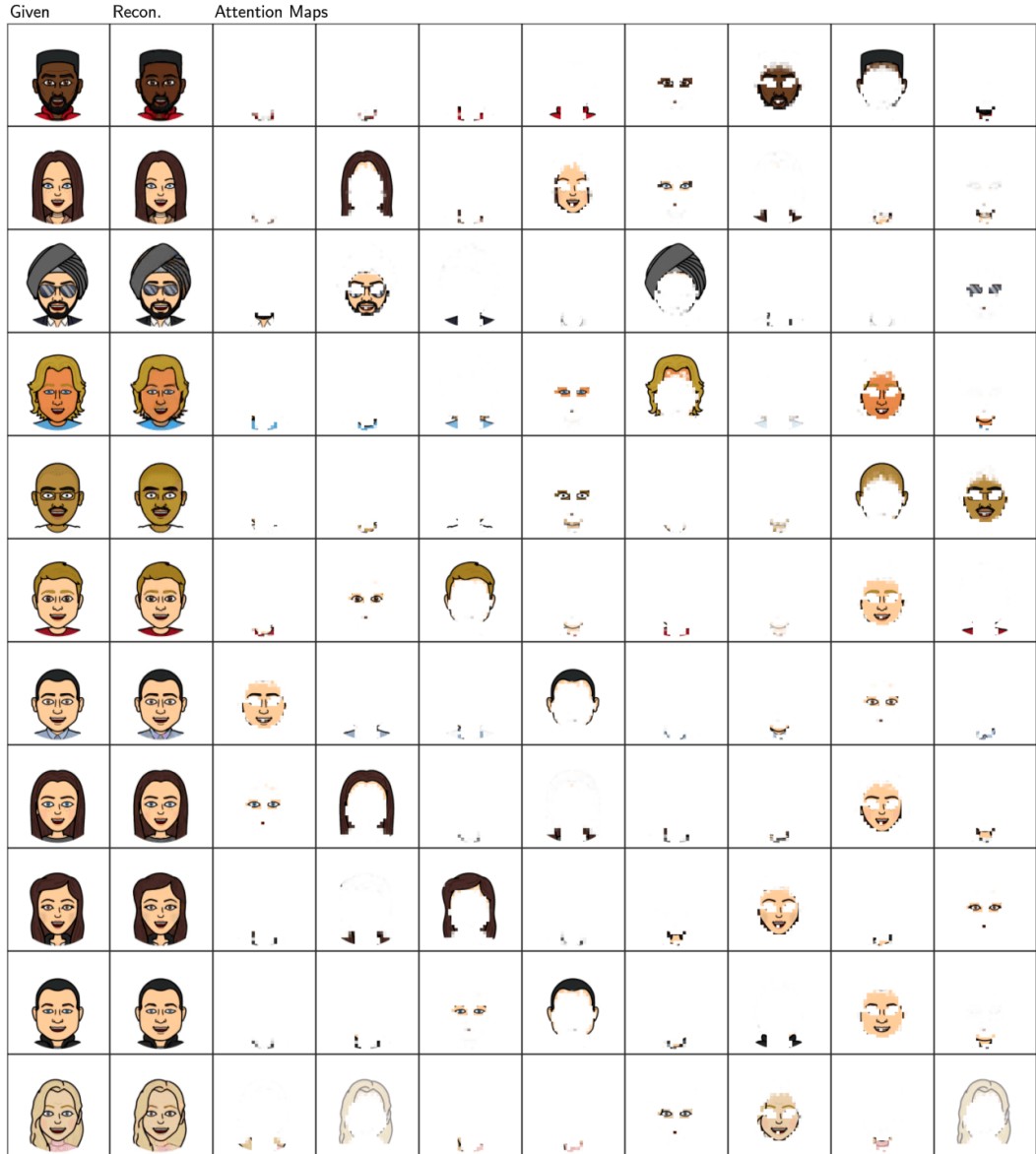

**Figure 26: Reconstruction and attention maps of slots in SLATE given the input images for Bitmoji.** We note that attention maps effectively localize on the individual meaningful segments of the faces.

## E.2 Object Discovery and Color Bias

Another problem that mixture decoders suffer from is what we term as the *color bias*. In mixture decoder, each object slot is commonly decoded using the spatial broadcast decoder to achieve a decoding capacity bottleneck. In this decoder, to decode each pixel, the pixel coordinate and the slot vector are concatenated and given to a decoder MLP to predict the pixel values for that coordinate. Hence, when two pixels are close to each other, the decoder MLP generates very similar pixel outputs for them. This is because the inputs to the MLP are very similar to each other for these two pixels. Given that an MLP, in general, prefers to learn smooth and linear (or constant) functions first before learning more complex functions, hence the output object rendered by a given slot tends to have a uniform or smoothly changing colors. As a consequence, this also biases the object-discovery process to also combine similar colors into a single slot and separate the regions with different colors into different slots.

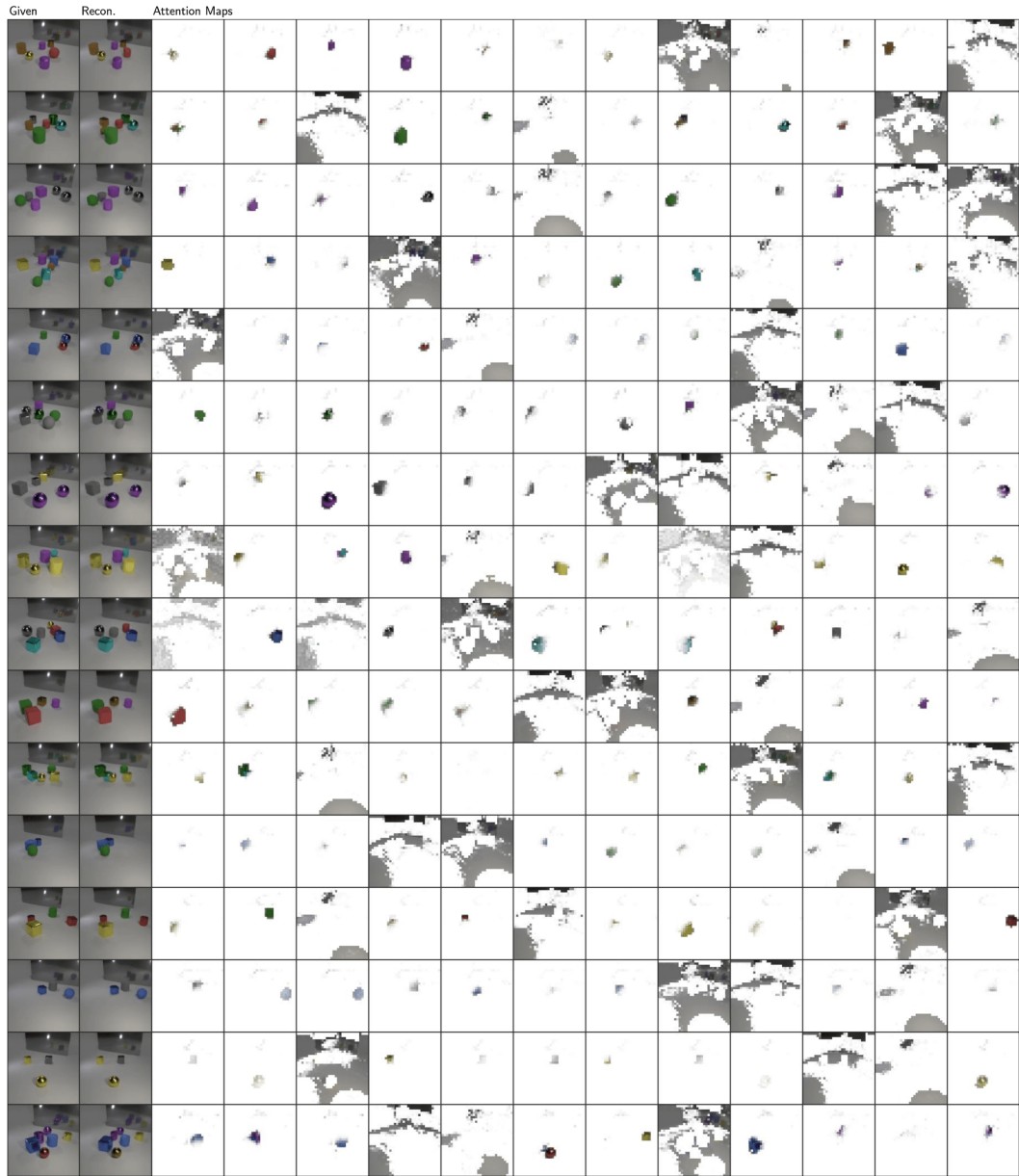

**Figure 27: Reconstruction and attention maps of slots in SLATE given the input images for CLEVR.** We note that attention maps effectively localize on the individual objects.

Consequently, this design creates the problem of *color bias* which arises in two forms when dealing with visually complex images: *i)* First, if a single object is made up of multiple colors, the object may get split into multiple slots to handle the complexity of rendering multiple colors. *ii)* Second, if there are multiple objects having a similar color, those objects can get merged and become represented as a single slot. In visually simple scenes having different objects with separate colors, this might not appear to be a problem. But in more natural scenes in which single objects can be made up of multiple colors and many objects can have similar colors, this becomes a problem and harms the quality of the discovered objects. This phenomenon is demonstrated in the object attention maps of mixture decoder in Figures 30, 32 and 31. In comparison, because our decoder is not biased towards simple or uniform colors, hence our object discovery process is more robust to the color bias issue.

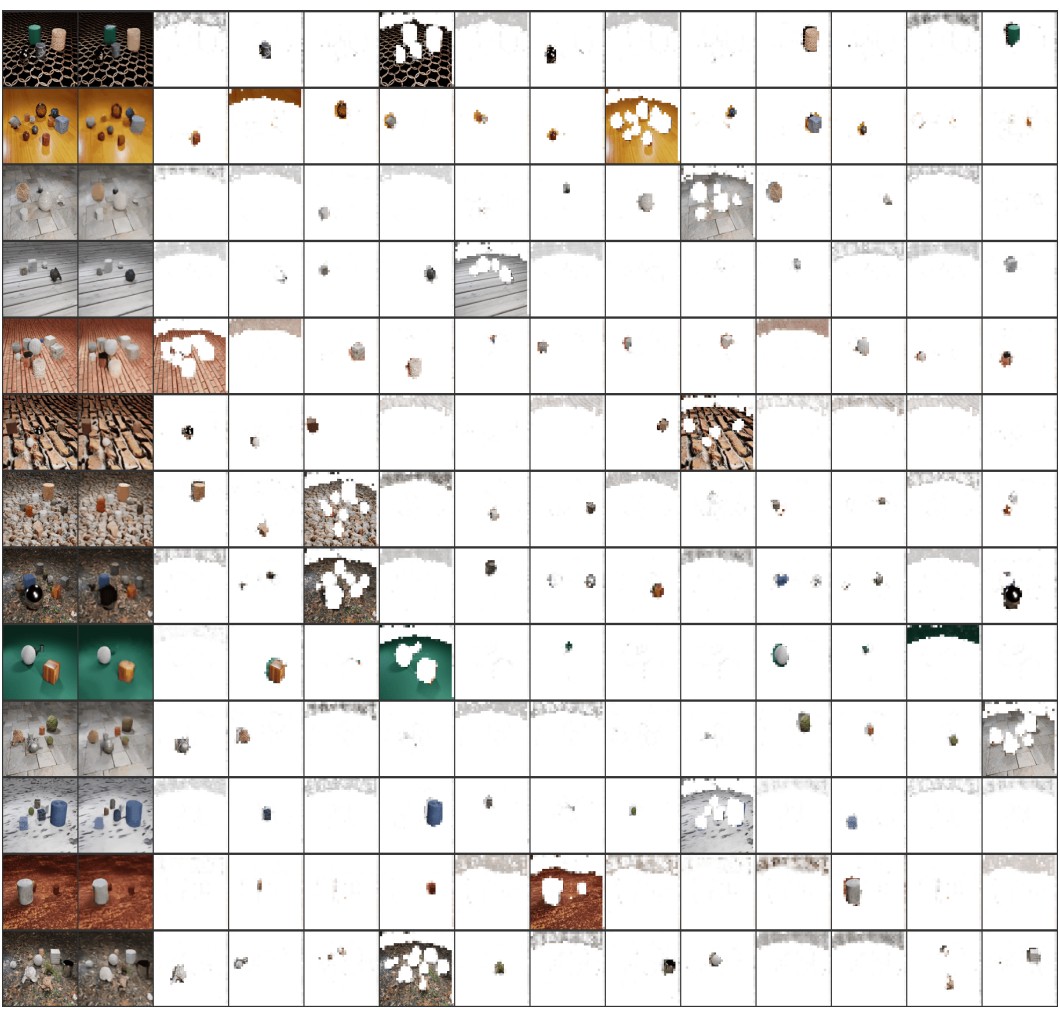

**Figure 28: Reconstruction and attention maps of slots in SLATE given the input images for CLEVRTex.** We note that attention maps effectively localize on the individual objects.

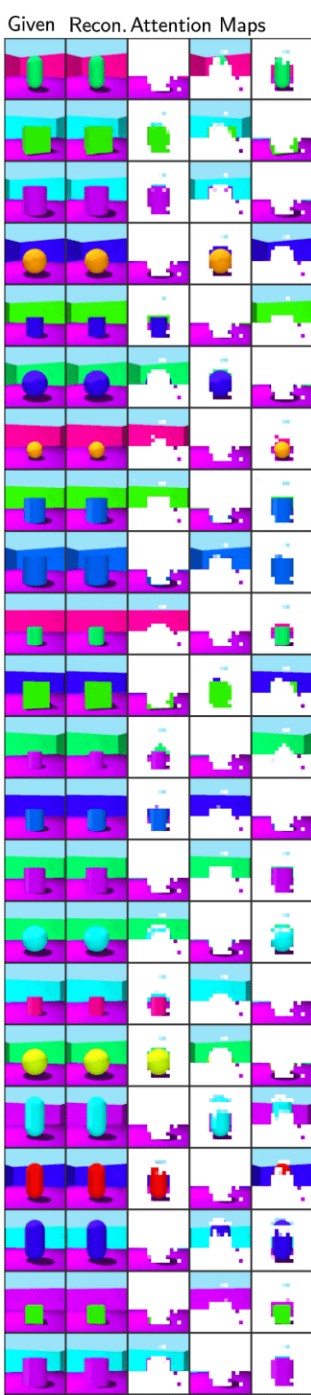

**Figure 29: Reconstruction and attention maps of slots in SLATE given the input images for 3D Shapes.**
We note that attention maps effectively localize on the individual meaningful segments such as wall, floor and the 3D object.

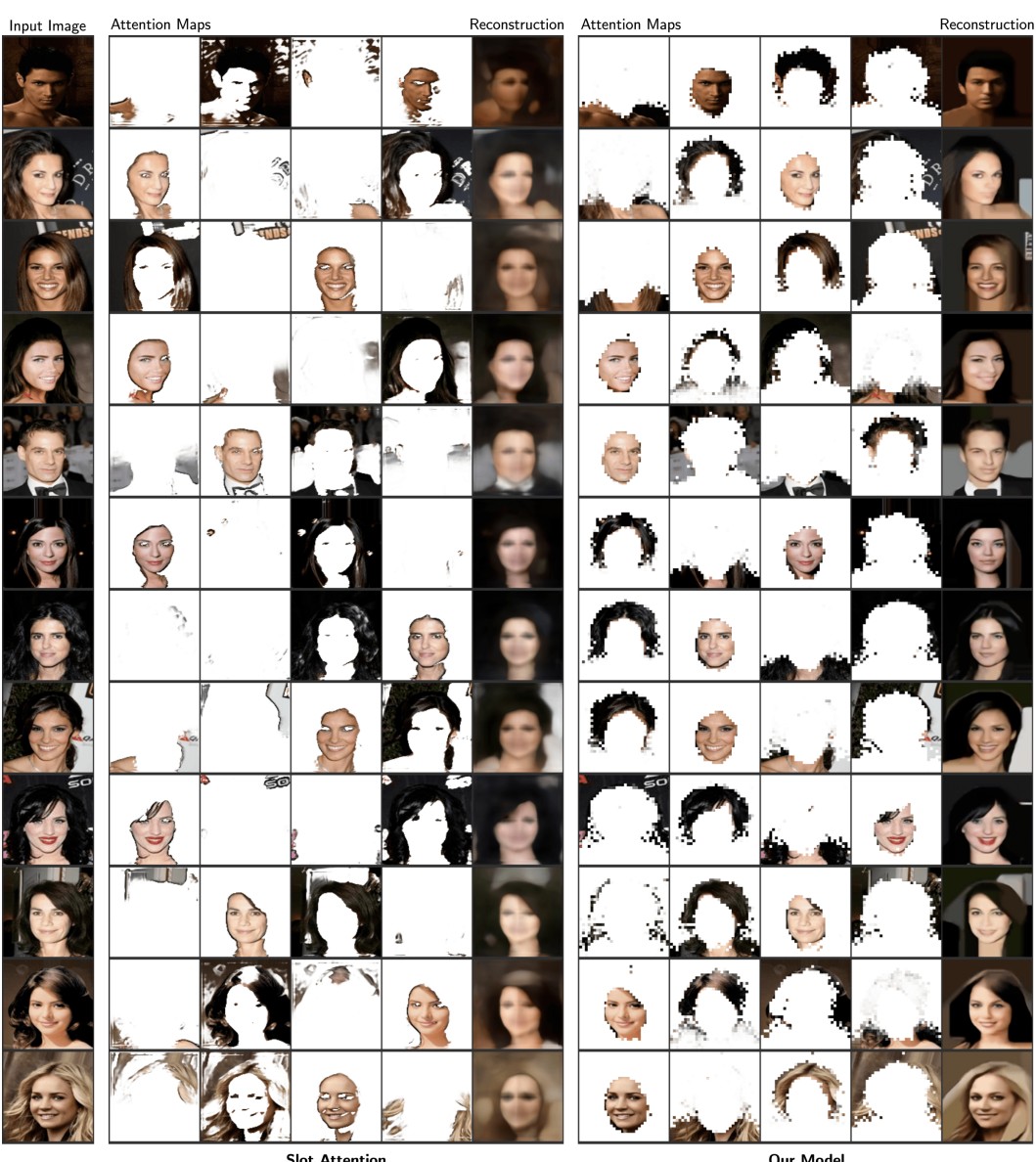

**Figure 30: Demonstration of Color Bias of Mixture-based Models in CelebA with Hard-to-Distinguish Background.** In this comparison, we consider the images in which the background has a very similar color as the hair color. We show the input image, the resulting object attention maps and the reconstruction generated by Slot Attention and by our model. In these, we note that due to the color bias problem, in Slot Attention, the hair and background have become merged into the same slot in most of the cases. However, in our model, this problem is resolved and we can see that the hair and the background are assigned to separate slots which is the desired decomposition. We also note how the mixture decoder produces blurry reconstructions while our reconstructions are significantly more clear.

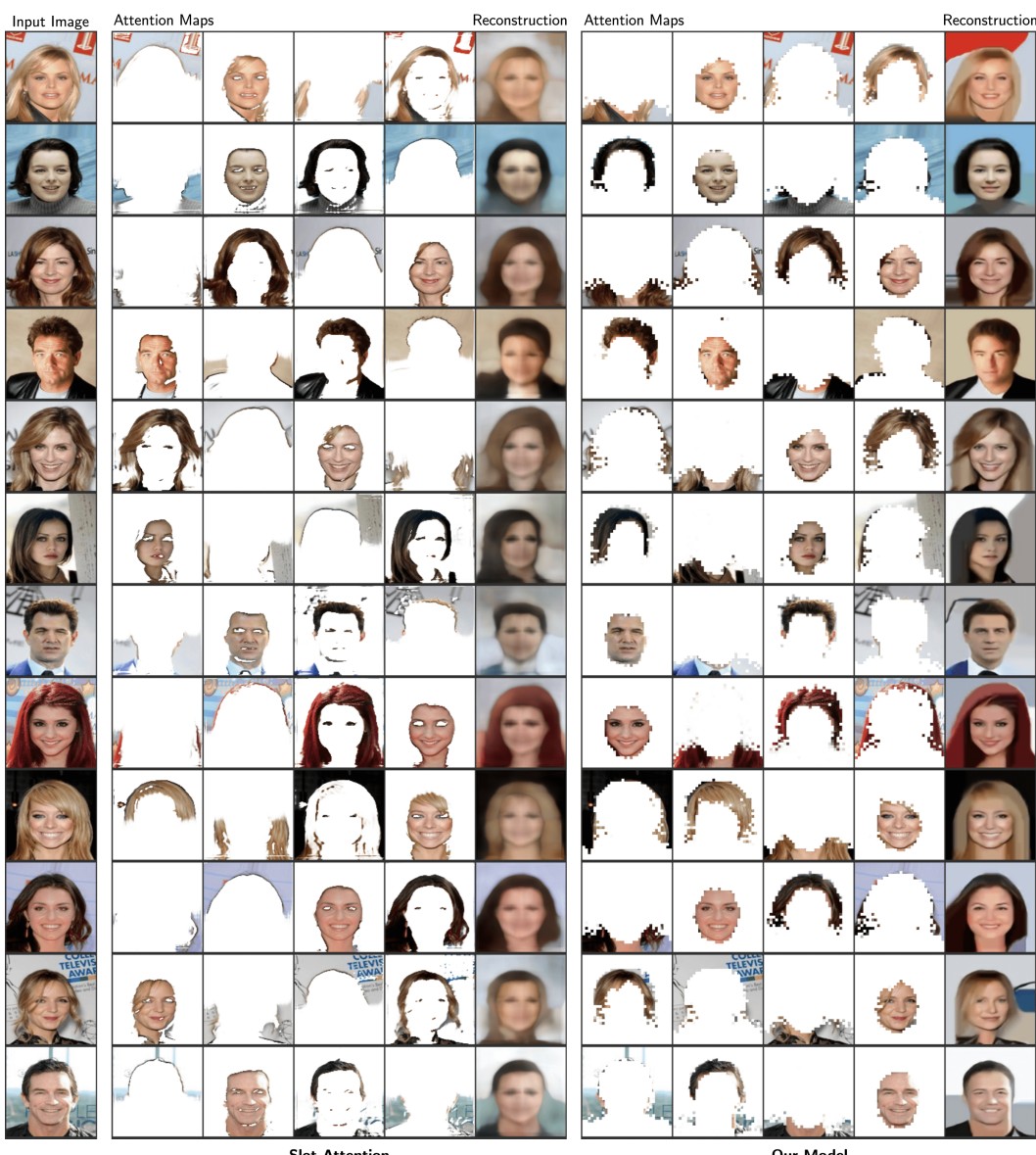

**Figure 31: Analysis of Color Bias of Mixture-based Models in CelebA with Easy-to-Distinguish Background.** We analyze the images in which the background has a color distinct from the hair color. This comparison is meant to show that mixture decoder may succeed when the colors of objects are distinct but when the colors are not distinct, it suffers as shown in Figure 30. Here, we show the input image, the resulting object attention maps and the reconstruction generated by Slot Attention and by our model. In these, we note that the color bias issue of mixture decoder is not that severe and both models can identify the hair slot as distinct from the background slot. However, we still observe that the mixture decoder produces blurry reconstructions while our reconstructions are significantly more clear.

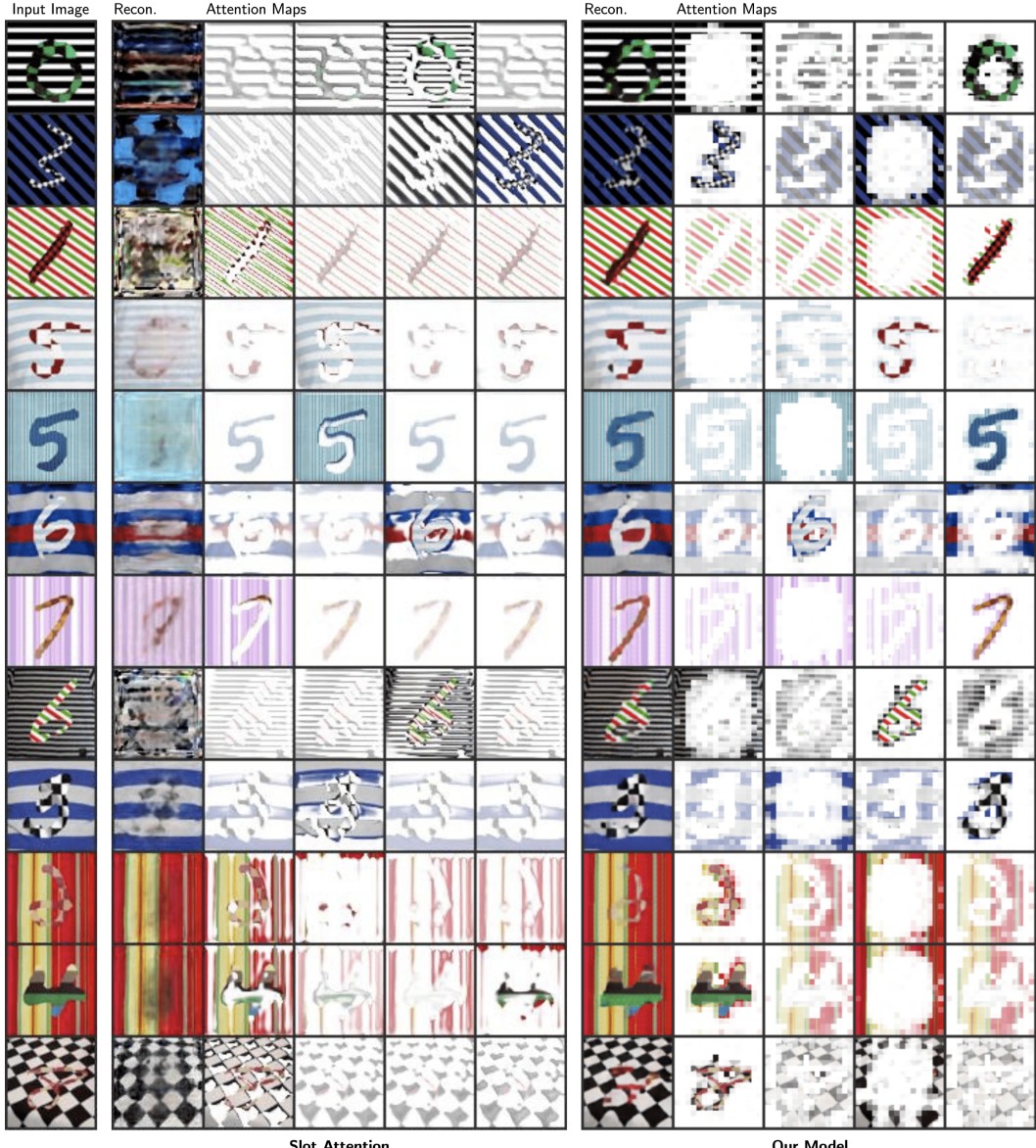

**Figure 32: Analysis of Color Bias of Mixture-based Models in Textured-MNIST.** We analyze the object attention maps of the mixture-based Slot Attention and those generated by our model. Here, we show the input image, the reconstruction and the resulting object attention maps produced by Slot Attention and by our model. In these, we note that Slot Attention suffers because of color bias as the parts of the digit and the parts of the background tend to get combined into a single slot. In contrast, in our model, the attention maps for the digits are clear and easily identifiable. Furthermore, our reconstructions are much more clear than those from the mixture-based decoder.

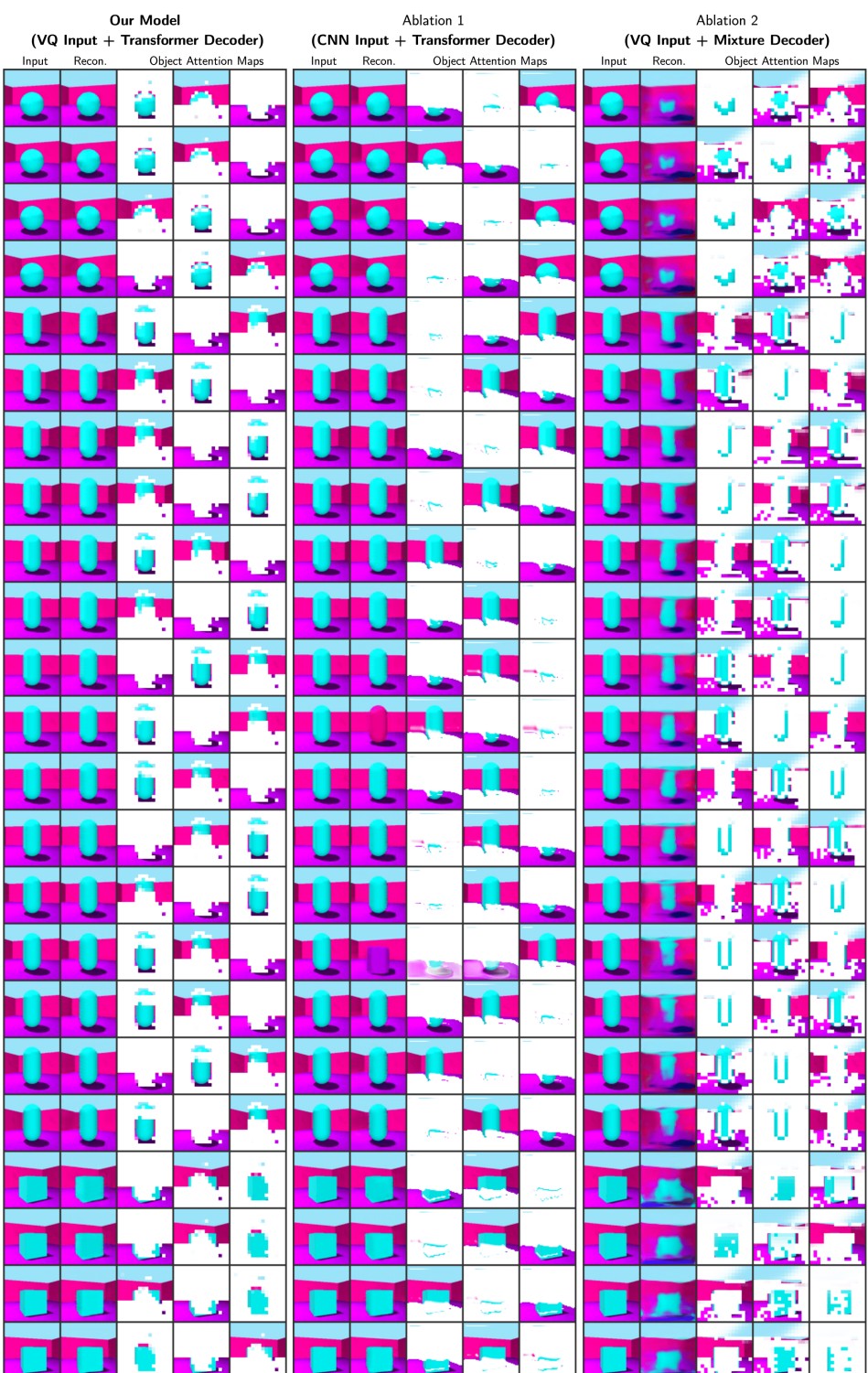

**Figure 33: Analysis of Object Attention Maps in Ablation Models.** We ablate our model by replacing the discrete token input (VQ) to the slot attention encoder with a CNN feature map as originally used by Locatello et al. (2020). We also ablate our model by replacing the Transformer decoder with the mixture decoder. These attention maps suggest that using DVAE for obtaining a discrete representation of the input image can be synergistic with Transformer decoder for achieving good object discovery.

## F  ADDITIONAL EXPERIMENT DETAILS

**Building Concept Library based on Object Position.** As described in Section 5.1, for CLEVR and Shapestacks, we sample object positions first and then sample an object for the chosen positions to build the slot prompts. Thus, in our concept library, we require the slots to be organized with respect to their positions. For this, we shall adopt a simple approach and using IOU between attention maps. By considering IOU, we are able to cluster together the object slots with similar patch regions regardless of the contents of the slots. To do this, we first construct a library of mask regions derived from a $G \times G$ grid. This is shown in Figure 34. For a given slot, we take its attention map and use it to assign to one of the mask regions in the library on the basis of high IOU. To build a slot prompt

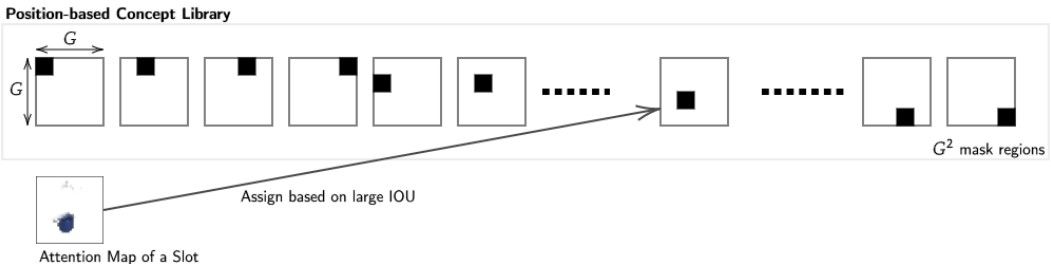

**Figure 34: Building a Position-based Concept Library.** We consider a set of mask regions derived from a $G \times G$ grid. For a given slot, we compute the IOU of the attention region of the slot with each region in the library and find the highest IOU match. We then assign this slot to this match region.

in CLEVR, we simply pick a set of regions from the library with some minimum clearance distance between them and then pick a slot for each of the picked regions. Similarly, in Shapestacks, we pick a given number of regions in a vertical tower configuration and for each region, we randomly sample an assigned slot. In this way, we construct the slot prompts for CLEVR and Shapestacks.

### F.1  CLEVR-MIRROR

In this work, we introduce a new dataset, CLEVR-Mirror as an extension of the standard CLEVR dataset. In comparison to standard CLEVR, in this dataset, we also introduce a mirror into the scene. CLEVR-Mirror is designed to see whether a model can obtain the ability to model complex global consistency by learning systematic relationship between local components. The standard CLEVR dataset is not proper for this purpose because the scene can easily be modeled without learning this global structure. CLEVR-Mirror requires learning relational knowledge such as the size change of reflected objects relative to the distance of the actual object to the mirror and occlusion among objects. For example, as shown in Figure 5, an object can be occluded in the mirror even if it is not in the non-mirror region.

### F.2  TEXTURED MNIST

We created this dataset using MNIST digits and textures from the Describable Textures Dataset (Cimpoi et al., 2014).

