# OpenReview forum: "Illiterate DALL-E Learns to Compose"
_ICLR.cc/2022/Conference — ICLR 2022 Poster_

### Official Review · Reviewer_8H9i · 2021-10-29

**Correctness:** 4
**Technical Novelty And Significance:** 2
**Empirical Novelty And Significance:** 3
**Recommendation:** 6
**Confidence:** 5

**Main Review:**

Strengths:
1. Overall, this paper is well written, and the technical details are easy to follow.
2. The main idea of learning compositional slot-based representations with DALLE regardless of text is interesting.
3. The experiment results strongly support the benefits of the object-centric approach.

Weaknesses:

**Synthetic Data.** As shown in multiple works [1,2,3], the Slot Attention and similar methods are currently limited to toy data like moving 2D objects or very simple 3D scenes and generally fail at more realistic data with complex textures. The authors proposed to modify DALLE, which is a zero-shot image generation model that works well on natural images, to work on synthetic datasets. It is hard for me to see how this approach generalizes to real and more complex data.

[1] Multi-object representation learning with iterative variational inference, ICML 19.

[2] Clevrtex: A texture-rich benchmark for unsupervised multi-object segmentation.

[3] Track, check, repeat: An EM approach to unsupervised tracking, arxiv 2021.


**Slots Initialization.** The authors mentioned the slot-decoding dilemma and the pixel-independence assumption that their model solves. But, as [1] and others suggested, using learnable query vectors instead of Gaussian-initialized slots helps the slots to learn on a unique embedding (similar to DETR). Thus, I am not sure the two mentioned problems above are related to the mixture decoder as the authors suggested but more to the slot encoding (and not the decoder). I would be happy to hear the author's thoughts about it.

[1] Self-supervised Video Object Segmentation by Motion Grouping, ICCV 2021


**Experiments.** The authors constantly compared to the Slot Attention model, but is it the correct comparison? Why not compare it to the DALLE, which does not use object-centric representations? Besides the fact that Slot Attention does not have an excellent image generator, the main point of this work is to show that adding compositional representations to DALLE improves the model (in contrast to adding a better decoder to the Slot Attention, unless I missed something).

Additionally, few other object-centric approaches use object-centric representations for image generation and manipulation, which the authors are not comparing at all. For example, Scene-Graph-to-image methods [1,2,3] aim to add compositionality representations for image generation. It should be at least discussed why the authors approach is better than using a more structured approach for image generation.

[1] Learning Canonical Representations for Scene Graph to Image Generation, ECCV 2020.

[2] Specifying object attributes and relations in interactive scene generation, ICCV 19.

[3] Image Generation from Scene Graphs, CVPR 18.


**Novelty Clarification.** I am somehow confused about the novelty of the paper. Do the authors want to emphasize that adding a more robust decoder framework from DALLE improves the Slot Attention model, or adding the object-centric representations enhances DALLE generation capabilities? I am not sure what the authors want to highlight, but the latter is the more interesting research question to me.


**Relation to Prior Work.** As mentioned earlier, few object-centric approaches try adding compositional representations for image/video synthesis and manipulation, which the authors are not discussing. I believe they could have been mentioned in the Related Section. I am writing a few below.

[1] Learning Canonical Representations for Scene Graph to Image Generation, ECCV 2020.

[2] Compositional Video Synthesis with Action Graphs, ICML 2021.


**DALLE.** The authors mentioned DALLE as a model which already has compositionality capabilities (See “However, from the perspective of compositionality, this success is somewhat expected because the text prompt already brings the composable structure. That is, the text is already discretized into a sequence of concept modules.”) I can't entirely agree with the authors, DALLE lacks compositionality as the results suggested by the authors (See the example “a stack of 3 cubes. a red cube is on the top, sitting on a green cube. the green cube is in the middle, sitting on a blue cube. the blue cube is on the bottom” in https://openai.com/blog/dall-e/). It seems the Slot Attention has much better compositionality capabilities than DALLE.






**Summary Of The Paper:**

This paper introduces Illiterate DALL·E, a zero-shot image generation model inspired by slot attention and DALLE without text. The main idea is to leverage object-centric representations into image generation. The method demonstrates the zero-shot generation of novel images without text and better quality in generation than the models based on mixture decoders.


**Summary Of The Review:**

My main concerns are that I cannot see how the proposed approach can generalize to more realistic domains, and I am confused about the main paper's story. Overall, I like this paper and its contribution, but I think the authors should highlight and show that their new model improves DALLE and not the Slot Attention model since DALLE could potentially benefit object-centric information for zero-shot image generation in real-life datasets. Furthermore, since DALLE has shown vulnerability to compositionality, I would expect the proposed author's approach to resolving it, and thus I feel this paper misses an excellent opportunity to do it.

I am open to the authors' feedback and other reviewers' opinions.


After Rebuttal
-----------------------------

After reading the authors' feedback and other reviewers' opinions, I would like to thank the authors for their rebuttal. The rebuttal addresses most of my concerns. I am leaning towards acceptance of the paper. I vote for 6.

---

> ### Author Response · Authors · 2021-11-23
> **Response to Reviewer 8H9i (1/3)**
>
> Thank you for reviewing our work!
>
> > ### The authors proposed to modify DALLE but did not evaluate on natural images.
>
>
> No, the goal of the proposed model is not to modify DALLE to outperform on natural image composition. Our model shows that even without text (a form of text-supervision in DALLE), we can obtain a model that still supports some degree of out-of-distribution compositionality. Even if one way to interpret our model is to see it as a text-**unsupervised** DALLE, the two models are fundamentally different. It is difficult to compare these two models because DALLE cannot generate without text and our model cannot generate with text-prompt. Even for non-direct comparison, we do not expect our text-**unsupervised** model to outperform DALLE (i.e., the text-supervised version) on natural image composition. That will be as difficult as to expect for an unsupervised segmentation method to outperform supervised segmentation methods. Rather, it is more accurate to see the goal of our proposed model as improving object-centric representation/generation models (to which DALLE does not belong to due to its lack of unsupervised representation learning from images). Due to the improved compositionality, one way to interpret our model is to see it as a text-unsupervised version of DALLE.
>
> Nevertheless, encouraged by the reviewer and the remarkable performance of our previous experiments, we tried to push the boundary with our model on two additional challenging datasets, CelebA and Textured-MNIST, and observed quite surprising results (we thank the reviewer for encouraging us to try this)
>
> Anonymous link to qualitative results for Textured-MNIST and CelebA is provided here: https://imgur.com/a/yfT0Mro. We visualize object attention maps, reconstructions, and compositional generations by combining arbitrarily drawn slot prompts.
>
> For textured-MNIST, we can observe from the Imgur-Figure 1 that our model successfully identifies foreground digits (the main object) from the complex and noisy background. Some digits are hard to identify even for humans (e.g., the digit 8 in the last row of the Imgur-Figure 1) but we can see our model successfully identifying it. On the contrary, slot attention fails in identifying digit-object slots by mixing the attention across both the foreground and the background. To our knowledge, in this setting, this is the first success in unsupervised object-centric representation learning. From Imgur-Figure 3 and 4, we also observe that the learned concepts and their compositions are successful. Below, we report the quantitative results for Textured-MNIST.
>
> | Metrics on Textured-MNIST   | Slot Attention   | Ours |
> | ---------------------------- | ---- | ---- |
> | Reconstruction MSE  | 822.73 | **149.02** |
> |  Reconstruction FID |  179.95    |  **55.00** |
> | Compositional Generation FID | 245.09 | **64.15** |
>
> For Celeb-A shown in Imgur-Figure 2, we observe that slot attention shows a tendency to group similarly colored pixels as a single object even if they are semantically different objects. For example, in many images, hair, eyes, background are all black, and slot attention groups these as an object. Surprisingly, we can see that our model semantically and correctly separates these into the three objects of hair, face, and background even when the boundary of hair and background is almost invisible. Also, the reconstruction quality of our model is much better than slot attention.
>
> For compositions of Celeb-A, we are still generating the compositions and computing the quantitative metrics. Due to the lack of time, we are not able to share them here, but we are looking forward to adding the results in the updated version.

---

> > ### Author Response · Authors · 2021-11-23
> > **Response to Reviewer 8H9i (2/3)**
> >
> > > ### Not sure the two mentioned problems (slot-decoding dilemma and the pixel-independence assumption) are related to the mixture-decoder or more to the slot encoding.
> >
> > This is an interesting point. We investigate the benefits of our architectural design choices by running additional ablations. In terms of architectural components, our model can be seen as VQ-Input + Slot Attention + Transformer Decoder while our baseline (Locatello et al.) can be seen as CNN Input + Slot Attention + Mixture Decoder. Hence components that differ between our model and the baseline are: VQ input layer and the Transformer decoder. We analyze the effect of these individual components by performing the following two ablations. In the first ablation, we take our model and replace the VQ input layer (i.e. DVAE) with a CNN encoder. We refer to this ablation as CNN Input + Transformer Decoder. In the second ablation, we take our model and replace the Transformer decoder with the mixture decoder. We refer to this ablation as VQ-Input + Mixture Decoder. For these ablations, we report below the FID score for the 3D Shapes dataset for the compositional generation task.
> >
> >
> >
> > | Model | Compositional Generation FID |
> > | -------- | -------- |
> > |  Transformer-based Model 1 (VQ Input + Transformer Decoder)        |  **36.75**        |
> > |  Transformer-based Model 2 (CNN Input + Transformer Decoder)        |  **32.07**        |
> > |  Mixture-based Model 1 (VQ Input + Mixture Decoder)        |   174.83       |
> > | Mixture-based Model 2 (CNN Input + Mixture Decoder)     |  44.14    |
> >
> > This result suggests that the transformer decoder is mainly responsible for gains in performance because the two transformer decoder models outperform the mixture decoder models. Interestingly, the CNN Input + Transformer seems slightly better than VQ input + Transformer in terms of FID. However, we found that the CNN Input + Transformer model has poorer object attention maps than VQ input as we show in this anonymous link: https://imgur.com/a/RYJ0sYw. This suggests that the design choice of using VQ input is also reasonable.
> >
> >
> > Although we performed this ablation study on the simple 3D shapes dataset due to the limited time, we are currently running the experiments for the other more challenging datasets as well. We expect the main result to not change on these datasets based on the fact that our model is shown to outperform the baseline more significantly on these more challenging datasets.
> >
> > > ### Why not compare to DALLE because the main point of this work is to show that adding compositional representations to DALLE improves the model.
> >
> > There seems to be some misunderstanding here. It is not true that we claim to add compositional representations to outperform DALLE. Hence we do not need to perform this comparison between the text-supervised DALLE with our text-unsupervised DALLE which is an unfair comparison as discussed in the previous answer.
> >
> > > ### The authors do not compare with generation-only models that do not use object-centric representations.
> >
> > The topic and scope of the paper are about models supporting both object-centric representation learning and compositional generation. Thus, we do not compare with models that only support generation. We discuss these generation-only models in the related work section and will update with your references.
> >
> > > ### It should be discussed why the author's approach is better than using a more structured approach for image generation, e.g.,
> > > - Learning Canonical Representations for Scene Graph to Image Generation, ECCV 2020.
> > > - Compositional Video Synthesis with Action Graphs, ICML 2021.
> > > - Learning Canonical Representations for Scene Graph to Image Generation, ECCV 2020.
> > > - Specifying object attributes and relations in interactive scene generation, ICCV 19.
> > > - Image Generation from Scene Graphs, CVPR 18.
> >
> > We updated the paper and discussed these models in the related works. However, all these mentioned papers are supervised models taking scene graph annotations as input whereas our model is an unsupervised object-centric representation learning model. The scope of our paper is different and the contributions of these papers are orthogonal to ours.
> >
> > > ### I am somehow confused about the novelty of the paper. Do the authors want to emphasize that adding a more robust decoder framework from DALLE improves the Slot Attention model, or adding the object-centric representations enhances DALLE generation capabilities? I am not sure what the authors want to highlight, but the latter is the more interesting research question to me.
> >
> > Our main contribution is the former. The key novelty of our model is to show that it endows a new important capability: out-of-distribution compositional generalization, which has not been achieved before in object-centric representation models. Thus, we believe our contribution is significant.

---

> > > ### Author Response · Authors · 2021-11-23
> > > **Response to Reviewer 8H9i (3/3)**
> > >
> > > > ### Discuss relation to prior works that do not use unsupervised compositional representation learning.
> > >
> > > We thank you for the reference. We have updated our related work section with the suggested discussion and reference.
> > >
> > > > ### DALLE still has some limitations in learning compositionality as it is shown that it cannot solve some compositional problems.
> > >
> > >
> > > We totally agree that the compositionality of DALL-E is not perfect. We did not claim or intend to claim that DALL-E provides perfect compositionality. The point of what we wrote is that the text input was crucial in making DALLE achieve the level of compositionality shown in the DALLE paper. We will clarify this.

---

> ### Author Response · Authors · 2021-12-06
> **Official Score Is Still Showing the Old Score**
>
> Dear Reviewer,
>
> We are encouraged by the upgrade to our score and the positive recommendation!
>
> We wanted to mention that the official score recommendation is still showing as 5 rather than the new score **6** as mentioned by you in the updated review. Hence, if possible, we would greatly appreciate it if the official score also reflects this change. Thank you and we are eager to hear any other thoughts and feedback!

---

> > ### Comment · Reviewer_8H9i · 2021-12-06
> > **The score is fixed**
> >
> > .

---

> > > ### Author Response · Authors · 2021-12-06
> > > **Thank You for the Fix**
> > >
> > > Dear Reviewer,
> > >
> > > Thank you for the fix!

---

### Official Review · Reviewer_KXKd · 2021-11-02

**Correctness:** 3
**Technical Novelty And Significance:** 3
**Empirical Novelty And Significance:** 3
**Recommendation:** 6
**Confidence:** 4

**Main Review:**

Strengths:

S1: The idea is inspiring and the visual prompts seem meaningful and interesting.
S2: The performance is fine based on the qualitative examples.

Weaknesses:

My major concern is that the experiments are simple. The datasets used in the paper are synthetic and easy to model. So I am wondering the performance on natural images, such as ImageNet, MSCOCO and Celeb1M, can the proposed model still learn meaningful prompts and what are the prompts for natural scenes? Another weakness is that the author does not conduct a human evaluation on the generated images, so it is difficult to judge how well the model performs.

**Summary Of The Paper:**

This paper proposes a  model that uses visual prompts to generate images. Basically, visual prompts are interesting and inspiring. And compared with existing works, the proposed model shows better performance.

**Summary Of The Review:**

Basically, the idea in the paper is interesting, but the experiments should be improved.

---

> ### Author Response · Authors · 2021-11-23
> **Response to Reviewer KXKd**
>
> Thank you for reviewing our work! We are encouraged by the positive recommendation!
>
> > ### Synthetic datasets are easy to model. Experiments on natural images are missing.
>
> Although the images are synthetic, they are not necessarily easy because the current state-of-the-art models suffer on those images. Thus, the line of research about unsupervised object-centric representation learning (including SPACE, IODINE, MoNET, Slot Attention, etc.) has focused mostly on such synthetic images (and also because it supports systematic control of experiments and analysis). To our knowledge, there is no model in this line yet that works reasonably on natural images---except some that, despite being natural, are as simple as the synthetic ones.
>
> Nevertheless, encouraged by the reviewer and the remarkable performance of our previous experiments, we tried to push the boundary with our model on two additional challenging datasets, CelebA and Textured-MNIST, and observed quite surprising results (we thank the reviewer for encouraging us to try this).
>
> Anonymous link to qualitative results for Textured-MNIST and CelebA is provided here: https://imgur.com/a/yfT0Mro. We visualize object attention maps, reconstructions, and compositional generations by combining arbitrarily drawn slot prompts.
>
> For textured-MNIST, we can observe from the Imgur-Figure 1 that our model successfully identifies foreground digits (the main object) from the complex and noisy background. Some digits are hard to identify even for humans (e.g., the digit 8 in the last row of the Imgur-Figure 1) but we can see our model successfully identifying it. On the contrary, slot attention fails in identifying digit-object slots by mixing the attention across both the foreground and the background. To our knowledge, in this setting, this is the first success in unsupervised object-centric representation learning. From Imgur-Figure 3 and 4, we also observe that the learned concepts and their compositions are successful. Below, we report the quantitative results for Textured-MNIST.
>
> | Metrics on Textured-MNIST   | Slot Attention   | Ours |
> | ---------------------------- | ---- | ---- |
> | Reconstruction MSE  | 822.73 | **149.02** |
> |  Reconstruction FID |  179.95    |  **55.00** |
> | Compositional Generation FID | 245.09 | **64.15** |
>
> For Celeb-A shown in Imgur-Figure 2, we observe that slot attention shows a tendency to group similarly colored pixels as a single object even if they are semantically different objects. For example, in many images, hair, eyes, background are all black, and slot attention groups these as an object. Surprisingly, we can see that our model semantically and correctly separates these into the three objects of hair, face, and background even when the boundary of hair and background is almost invisible. Also, the reconstruction quality of our model is much better than slot attention.
>
> For compositions of Celeb-A, we are still generating the compositions and computing the quantitative metrics. Due to the lack of time, we are not able to share them here, but we are looking forward to adding the results in the updated version.
>
>
> > ### Need for Human Evaluation
>
>
> Thank you for the suggestion! We performed a human evaluation and we have updated the results in the paper. We presented 6 human participants each with 100 pairs of compositional generations generated using Slot Attention and our model. Below, we show the percentage votes received in favor of our model over Slot Attention. We can see that the generations of our model were preferred over the generations from the Slot Attention except for 3DShapes which is almost the same. Although the number of participants is not large, the large gap in the preference rate makes the result meaningful. We hope to update the results with more participants in the final version.
>
> | Dataset | Percentage of votes in favor of our model |
> | -------- | -------- |
> | 3DShapes |   50.36%       |
> | CLEVR-Mirror |  84.66%  |
> | Shapestacks |  78.00%    |
> | Bitmoji | 95.06%     |

---

> ### Author Response · Authors · 2021-12-06
> **Update After Reading Our Rebuttal?**
>
> Dear Reviewer,
>
> We are eager to hear your further thoughts after reading our rebuttal. As suggested by you, we have provided additional results on visually more challenging datasets (Textured-MNIST and CelebA) and also added a human evaluation. Do feel free to mention any further concerns or questions to us. Thank you!

---

### Official Review · Reviewer_o6Nv · 2021-11-02

**Correctness:** 3
**Technical Novelty And Significance:** 2
**Empirical Novelty And Significance:** 3
**Recommendation:** 6
**Confidence:** 4

**Main Review:**

STRENGTHS
- Identifies an area of opportunity between DALL-E (which has strong generation and imagination abilities, conditioned on explicit concepts) and Slot learning models (which can recover images based on slots but have limited imaginative ability) and addresses the gap with an image composition model
- Good exploration of the weaknesses of pixel-mixture decoders for image generation from slots, including its inability to leverage strong decoders and independence / lack of dependence between slots/concepts.
- The model demonstrates promise in out-of-distribution composition, and the experimental setup there is well founded.

WEAKNESSES
- Lacks detail on why CLEVR-Mirror was mirrored from CLEVR and how/why it was used in place of CLEVR.
- Would appreciate more clarity on architectural / modeling choices, such as the decision to use DVAE over VQ-VAE (which is used for ImageGPT).
- Stemming from above, this paper could also use a discussion of modeling ablations or comparisons to assess whether the slot2seq approach as a whole is responsible for the empirical gains in generation quality or if there is a strong dependence on architecture.
- It would be useful to discuss or experiment with imposing an order on the concept/slot prompt, as there may be an existing hierarchy when imagining an image (similar to the somewhat unspoken ordering exists for adjective ordering in English)
- Would like to see more than just comparison between mixture decoders and Slot2Seq - for example, [1] introduces energy-based models for composition, which seem to be applicable in this setting and would serve as a good point for contextualizing the overall impact of this work.


Additional References to include:
[1] Du, Yilun, Shuang Li, and Igor Mordatch. "Compositional visual generation with energy based models." Advances in Neural Information Processing Systems 33 (2020): 6637-6647.

**Summary Of The Paper:**

This paper proposes a method (Slot2Seq) to adapt the recent DALL-E (text-to-image) model to perform image-to-image composition. The aim is to simultaneously learn latent concepts from base images that can then apply to the generation process (as opposed to input text, which contains somewhat discretized "concepts" already). Experiments with 4 datasets (including a mirrored version of CLEVR) demonstrate that the Slot2Seq approach is effective for novel image generation from slots, reconstruction, and out-of-distribution generation.

**Summary Of The Review:**

The paper introduces a simple way to perform image composition using slot attention and a GPT-based decoder in place of pixel-mixture decoders. The ablation study is promising, but I would like to see the inclusion of additional baselines and/or more ablation studies to properly contextualize this work.

---

> ### Author Response · Authors · 2021-11-23
> **Response to Reviewer o6Nv (1/2)**
>
> Thank you for the insightful suggestions and the feedback!
>
> > ### Lacks detail on why CLEVR-Mirror was mirrored from CLEVR and how/why it was used in place of CLEVR.
>
> We agree about the need for clarifying the motivation of the dataset. We added the following discussion in the updated version.
>
> CLEVR-Mirror is designed to see whether a model can obtain the ability to model global consistency by learning complex systematic relationships between local components. For example, in CLEVR-Mirror, the model must obtain knowledge about the size change of reflected objects relative to the distance of the actual object to the mirror and occlusion among objects. As shown in Figure 5, an object can be occluded in the mirror even if it is not occluded in the non-mirror region. Standard CLEVR cannot serve this purpose because the scene can easily be modeled without learning this systematic global relationship structure.
>
> > ### More clarity on architectural / modeling choices, such as the decision to use DVAE over VQ-VAE (which is used for ImageGPT).
>
> In the updated version, we clarified the motivations of our architectural design choices. DVAE is chosen because DALLE-E also used it and because it is also simpler than VQ-VAE. Because unlike VAE our model does not need to generate from prior, removing the KL term is acceptable and we observe the effect of doing this is negligible while we can enjoy the simplicity and training efficiency from it.
>
> > ### Discuss modeling ablations or comparisons to assess whether the slot2seq approach as a whole is responsible for the empirical gains in generation quality or if there is a strong dependence on architecture.
>
> Thank you for this suggestion. To investigate this more clearly we performed additional experiments. We have also added these to the paper.
>
> In terms of architectural components, our model can be seen as VQ-Input + Slot Attention + Transformer Decoder while our baseline (Locatello et al.) can be seen as CNN Input + Slot Attention + Mixture Decoder. Hence components that differ between our model and the baseline are: VQ input layer and the Transformer decoder. We analyze the effect of these individual components by performing the following two ablations. In the first ablation, we take our model and replace the VQ input layer (i.e. DVAE) with a CNN encoder. We refer to this ablation as CNN Input + Transformer Decoder. In the second ablation, we take our model and replace the Transformer decoder with the mixture decoder. We refer to this ablation as VQ-Input + Mixture Decoder. For these ablations, we report below the FID score for the 3D Shapes dataset for the compositional generation task.
>
> | Model | Compositional Generation FID |
> | -------- | -------- |
> |  Transformer-based Model 1 (VQ Input + Transformer Decoder)        |  **36.75**        |
> |  Transformer-based Model 2 (CNN Input + Transformer Decoder)        |  **32.07**        |
> |  Mixture-based Model 1 (VQ Input + Mixture Decoder)        |   174.83       |
> | Mixture-based Model 2 (CNN Input + Mixture Decoder)     |  44.14    |
>
> This result suggests that the transformer decoder is mainly responsible for gains in performance because the two transformer decoder models outperform the mixture decoder models. Interestingly, the CNN Input + Transformer seems slightly better than VQ input + Transformer in terms of FID. However, we found that the CNN Input + Transformer model has poorer object attention maps than VQ input as we show in this anonymous link: https://imgur.com/a/RYJ0sYw. This suggests that the design choice of using VQ input is also reasonable.
>
> Although we performed this ablation study on the simple 3D shapes dataset due to the limited time, we are currently running the experiments for the other more challenging datasets as well. We expect the main result to not change on these datasets based on the fact that our model is shown to outperform the baseline more significantly on these more challenging datasets.

---

> > ### Author Response · Authors · 2021-11-23
> > **Response to Reviewer o6Nv (2/2)**
> >
> > > ### Discuss or experiment with imposing an order on the concept/slot prompt
> >
> > Thank you for suggesting this interesting idea. Models like MoNet and GENESIS apply sequential attention at the object-level and we may interpret this attention order as such a structure. Although slot attention, which does not consider an order among the slots, outperforms these models, we agree that it is still worth investigating further about the potential of discovering and utilizing some latent geometric structure among slots. Perhaps, the multiple layers of the transformer might already discover such structure implicitly. As this requires a significant change in the model, we will discuss it as a future work.
> >
> > > ### More ablation studies and additional baselines such as [Du, Yilun, et. al.].
> >
> > We performed additional ablation studies comparing different architectural design choices. This is already discussed in the previous answer above.
> >
> > About more baselines, it is unclear to which additional baselines we need to compare additionally. The suggested model [Du, Yilun, et. al.] lacks the ability to decompose a given image into objects **without supervision**. As the scope of our paper is about an unsupervised method, it seems not necessary to compare to this model. We have added the reference to this work in the updated paper.
> >
> > In this line of research, there are also other major unsupervised object-centric representation models such as MoNet, IODINE, and GENESIS to which we may compare. However, all these models use the mixture-decoder and Slot Attention is the state-of-the-art that outperformed the other models. We actually tested MoNET as well but observed it to be significantly inferior to Slot Attention and thus to ours. As discussed in the related works section, there are also GAN-based object-centric generation models. However, to our knowledge, we do not know any GAN model (not using a mixture decoder) that can **infer** object-centric representations from images.

---

> > > ### Comment · Reviewer_o6Nv · 2021-12-06
> > > **Thank you for addressing my concerns**
> > >
> > > These responses are very helpful to address the concerns I raised in my review. It's interesting to see such a drastic difference in the attention maps between using CNN and VQ inputs, despite the improvement in FID - this perhaps merits further experimentation and may gain more clarity in experiments on other datasets.
> > >
> > > It would be helpful to include the MoNET numbers as well to quantify what "significantly inferior" means in this case.
> > >
> > > I am revising my score to 6 in context of the rebuttal and the other reviews/rebuttals.

---

> ### Author Response · Authors · 2021-12-06
> **Any Updates After Reading Our Rebuttal?**
>
> Dear Reviewer,
>
> We believe we have addressed most of your concerns and provided the requested clarifications in the rebuttal and the updated paper. We are eager to hear your further feedback. If there are any other concerns or questions, do feel free to mention them to us. Thank you!

---

### Public Comment · ~Michael_Chang1 · 2021-11-10
**KL Loss of Discrete VAE**

Very interesting work!

What is the motivation for not using the KL loss from DALL-E in the discrete VAE?

Also, it seems like the (Im, et al. 2017) citation is not the correct one for the discrete VAE. Would the authors be able to clarify?

---

> ### Author Response · Authors · 2021-11-11
> **Response about the KL Loss of Discrete VAE**
>
> Thank you for the comment!
>
> > What is the motivation for not using the KL loss from DALL-E in the discrete VAE?
>
> Our model also works with a KL term but we found it more stable to use a small $\beta$ coefficient for the KL term (e.g. $\beta=0.04$ or smaller). The performance did not change much when we set $\beta=0$ which becomes the same as not using a KL term.
>
> > Also, it seems like the (Im, et al. 2017) citation is not the correct one for the discrete VAE.
>
> Thank you for pointing this out! This will be fixed in the updated version.

---

### Author Response · Authors · 2021-11-23
**Paper Updates**

We thank all the reviewers for taking the time to review our work and providing valuable feedback for improving the paper. Taking them into account, we have made the following updates to the paper. We will also provide a more detailed discussion to each reviewer.

1. Added results for two new datasets, Textured MNIST and CelebA, to show the benefits of our model on visually complex and natural images. For Textured-MNIST, we added the FID score for compositional generation, and MSE and the FID score for reconstruction. We also added visualizations of object discovery, concept library and compositional generations. For CelebA, we added visualizations of object discovery.
2. Added a human evaluation of compositionally generated images.
3. Added an ablation study using the 3D Shapes dataset and visualized the attention maps of the ablation models to better justify our design choices.
4. Updated the reconstruction metrics for the 3D Shapes dataset (The first row of Table 1. (b)).
5. Discussed the motivation for choosing CLEVR-MIRROR over CLEVR.
6. Discussed the motivation for choosing DVAE over VQ-VAE.
7. Discussed the motivation for not requiring a KL term.
8. Discussed additional suggested references in the related work.

There are some more experiments that we are still running and we look forward to adding them in the earliest revision.

---

### Decision · Program_Chairs · 2022-01-20

**Decision:**

Accept (Poster)

**Comment:**

The paper addresses the problem of generating images by combining visual components. These components are learned during pretraining, forming a dictionary of visual concpets which plays the role of text in DALLE. The technique is based on DALLE and slot attention approach to generate VQ codes in a way that is consistent.

Reviewers had various concerns, including (1) that using synthetic images makes it easier to combine visual components' (2) that the novelty and relation to literature was not clear enough (3) missing ablations.  The authors provided a detailed rebuttal which addressed reviewer concerns in a convincing way.

One remaining issue of the paper is the writing. The paper fails to clearly explain the workflow (what are input and output during pretraining, training and inference), and how compositionality is controlled (what can be used for conditioning). As a consequence, it requires substantial effort to understand the idea of the paper, and what real problems can be solved with the proposed approach .

The paper can be accepted to ICLR, but it is expected that the writing would be improved. The abstract and introduction should make concrete statements about what the approach does, what problems it solves and how it can be used for the various tasks as disucussed in the experiments